# Spatially adaptive estimation of multi-layer soil temperature at a daily time-step across China during 2010-2020

Xuetong Wang[1, 2], Liang He[3, *], Peng Li[1, 2], Jiageng Ma[4], Yu Shi[5], Qi Tian[1, 2], Gang Zhao[2, 6], Jianqiang He[7], Hao Feng[2], Hao Shi[8, 9, *], Qiang Yu[2, *]

[1] College of Natural Resources and Environment, Northwest A&F University, Yangling 712100, China

[2] State Key Laboratory of Soil and Water Conservation and Desertification Control, Northwest A&F University, Yangling 712100, China

[3] National Meteorological Center, Beijing, 100081, China

[4] Key Laboratory of Ecosystem Network Observation and Modeling, Institute of Geographic Sciences and Natural Resources Research, Chinese Academy of Sciences, Beijing 100101, PR China

[5] Institute of Carbon Neutrality, Sino-French Institute for Earth System Science, College of Urban and Environmental Sciences, Peking University, Beijing 100871, China

[6] College of Soil and Water Conservation Science and Engineering, Northwest A&F University, Yangling, Shaanxi, 712100, China

[7] Key Laboratory for Agricultural Soil and Water Engineering in Arid Area of Ministry of Education, Northwest A&F University, Yangling 712100, China

[8] State Key Laboratory for Ecological Security of Regions and Cities, Research Center for Eco-Environmental Sciences, Chinese Academy of Sciences, Beijing, 100085, China

[9] College of Resources and Environment, University of Chinese Academy of Sciences, Beijing, 100049, China

**Correspondence:**

Liang He (heliang_hello@163.com)

Hao Shi (haoshi@rcees.ac.cn)

Qiang Yu (yuq@nwafu.edu.cn)

**Abstract**

Soil temperature ($T_s$) is critical in regulating agricultural production, ecosystem functions, hydrological cycling and climate dynamics. However, the inherent spatial and temporal heterogeneity of soil thermal regimes constitutes a persistent challenge in obtaining high-resolution, continuous gridded $T_s$ datasets along vertical profiles. To address this issue, we propose a spatially adaptive layer-cascading Extreme Gradient Boosting (XGBoost) algorithm to generate daily multi-layer $T_s$ data (0, 5, 10, 15, 20, and 40 cm) at a spatial resolution of 1 km in China from 2010 to 2020. The methodology dynamically partitions non-uniformly distributed measuring sites (2,093 sites across the country) to quadtrees and incorporates thermal coupling effects propagated between neighbor soil layers. Multi-source data, including satellite retrievals of land surface temperature and vegetation index, and ERA5 reanalysis climate variables were used as inputs. Validation using both spatially independent test sets and flux-tower observations demonstrated the robustness and accuracy of the product. It is noted the model's performance was lower in summers and winters than in springs and autumns. Compared to existing global or regional $T_s$ products, the dataset developed here is characterized by its fine spatio-temporal patterns and high reliability, enabling it to provide supports for precision agriculture, ecosystem modeling and understanding climate-land feedback. Free access to the dataset can be found at https://doi.org/10.11888/Terre.tpdc.302333 (Wang et al., 2025b).

**Key words:** Soil temperature, spatially adaptive, machine learning, multi-source data

## 1. Introduction

Soil temperature ($T_s$) is a critical driver of ecosystem dynamics, influencing nearly all physical, chemical, and biological processes (Bayatvarkeshi et al., 2021; Xu et al., 2023; Liu et al., 2025). $T_s$ plays a pivotal role in land-atmosphere exchanges. By controlling the partitioning of net radiation into sensible and latent heat fluxes, $T_s$ directly shapes atmospheric boundary layer circulation, with cascading effects on regional climate patterns (Mahanama et al., 2008; Chen et al., 2021a). $T_s$ also drives soil freeze-thaw cycles, which are critical for hydrological processes in cold regions. Permafrost thaw alters subsurface water storage, runoff dynamics and groundwater recharge, with implications for both local and basin-scale hydrology (Zhang et al., 2005; Shati et al., 2018). In addition, it governs the rates of soil microbial activities, nutrient cycling, and organic matter decomposition, with direct implications for carbon dynamics. For instance, $T_s$ modulates microbial respiration, thereby regulating the release of organic carbon into the atmosphere as $CO_2$ that is central to global carbon cycling (Yang et al., 2011). Given its multifaceted influences on carbon cycling, climate feedbacks and hydrological systems, accurate $T_s$ estimation is indispensable for advancing ecosystem monitoring, refining climate models, and developing effective strategies to mitigate and adapt to climate change.

$T_s$ exhibits high heterogeneity at large spatial scales due to varying driving factors. Solar radiation changes its radiation intensity by adjusting the incident angle and sunshine duration, thus affecting the heating effects on surface soils (Wang and Dickinson, 2013). Additionally, diurnal variations of air temperature cause periodic changes in surface temperature, while the amplitude is often closely related to the local climate and topography. Furthermore, surface covers (e.g., vegetation and snow) significantly impact $T_s$ (Xu et al., 2020; Mortier et al., 2024). Vegetation canopies effectively intercept and scatter solar radiation, while root systems modulate soil moisture distribution, thereby stabilizing deeper soil temperatures (Li et al., 2024). Snow cover, characterized by high albedo, reflects substantial solar radiation and acts as an effective insulator, mitigating cold air penetration and maintaining warmer soil

temperatures during winter months (Myers-Smith et al., 2015). Moreover, thermal conductivity and heat capacity are critical parameters controlling vertical heat transfer in soils. Sandy soils have higher porosity and lower water retention, resulting in lower heat capacity and higher thermal conductivity, thus responding rapidly to temperature changes. In contrast, clay soils have lower porosity and stronger water retention, leading to higher heat capacity and significant thermal stability, characterized by delayed responses to temperature variations (Ochsner et al., 2001; Zhao et al., 2022). Understanding these mechanisms is essential for developing refined vertical $T_s$ distribution models and improving the accuracy of $T_s$ estimation.

Given these complex processes, accurately estimating $T_s$ across different depths is challenging. Quite a few models have been proposed for $T_s$ estimation. These models can be generally classified into physical, statistical or empirical, and machine learning (ML) types (Li et al., 2024; Farhangmehr et al., 2025). Physical models, derived from fundamental heat conduction laws and energy balance equations, provide explicit mechanistic interpretations but suffer from computational complexity and heavy reliance on multi-domain input parameters, which range from soil properties to climatic variables (Gao et al., 2008; Hu et al., 2016; Badache et al., 2016). Statistical or empirical models, such as autoregressive integrated moving average and regression methods (Xing et al., 2018), are usually limited to localized, small-sample applications. Data-driven ML techniques demonstrate a superior ability to capture nonlinear relationships and thus usually can obtain high prediction accuracy. For instance, at site scale, Feng et al. (2019) estimated multi-layer $T_s$ at half-hourly resolutions using Extreme Learning Machine, with a RMSE ranging from 2.26~2.95 K. Li et al., (2022) implemented an attention-aware long short-term memory (LSTM) model for predicting next-day $T_s$ and the model obtained a RMSE of 0.74~2.53 K. At the regional scale, Xu et al. (2023) integrated satellite remote sensing with a deep belief network model to reconstruct continuous $T_s$ profiles (at depths of 5–40 cm) across the Qinghai-Tibetan Plateau (QTP), obtaining $R^2 > 0.836$ and MAE < 2.152 °C. Similarly, Farhangmehr et al. (2025) developed a hybrid convolutional neural network-LSTM (CNN-LSTM) architecture for

predicting $T_s$ across North American climatic zones at 0~7 cm depths, with $R^2$ ranging from 0.93 to 0.99.

Although significant advances have been made in estimating $T_s$, large-scale $T_s$ prediction continues to confront critical challenges, sourcing from environmental complexity and methodological limitations. First, $T_s$ exhibits considerable spatial heterogeneity driven by regional disparities in topography, soil composition, vegetation density, and microclimate (Bayatvarkeshi et al., 2021). These factors create

nonstationary relationships between $T_s$ and explanatory variables (e.g., air temperature, soil moisture), necessitating regionally tailored modeling approaches. Second, data scarcity and uneven spatial distribution of site measurements introduce further complexity. Aggregating sparse, unevenly distributed measurements into a single model often leads to overfitting: high accuracy on training data but poor generalization to

underrepresented regions or previously unseen data (Li et al., 2024). Ultimately, developing models that reconcile scalability (for large spatial scales) with localized precision (to capture site-specific interactions) remains an unresolved priority, underscoring the persistent challenge of balancing universal applicability with spatially adaptive fidelity in $T_s$ prediction methodology.

Recent advances in spatially adaptive modeling have increasingly emphasized the importance of addressing spatial heterogeneity and uneven sampling density in environmental datasets. Classical quadtree structures and related hierarchical spatial data models provide the theoretical foundation for constructing adaptive, variable-sized spatial partitions, enabling efficient organization of multiscale spatial information

through recursive subdivision (Samet, 1984). Building on this foundation, Lagonigro et al., (2020) developed the AQuadtree R package, which provides an adaptive spatial partitioning framework capable of generating variable-sized grid cells according to the spatial distribution of observations. This adaptive partitioning produces finer grids in data-dense regions and coarser grids where observations are sparse, ensuring a spatial

structure that better reflects sampling heterogeneity and improves the model's capacity to capture localized spatial variability. Extending this idea, we develop a rotated-

quadtree strategy that applies multiple orientation angles during the quadtree subdivision process. This enhancement allows the model to capture spatial heterogeneity from multiple directional perspectives, and averaging predictions across rotation angles substantially reduces the boundary artifacts that may arise from single-angle grid partitioning, ultimately improving the robustness of local modeling under complex environmental gradients.

To address the irregular station distribution, and non-stationarity commonly encountered in large-scale $T_s$ estimation, we construct a spatially adaptive modeling framework based on the rotated quadtree approach. Within each grid cell, multi-source environmental predictors are integrated with in situ station records, and $T_s$ is estimated using XGBoost models. Based on this framework, the objectives of this study are to: (1) construct a spatially adaptive modeling system; (2) generate a multi-layer $T_s$ dataset at a daily time-step and one kilometer resolution in China from 2010-2020; and (3) evaluate the dataset through independent validation with flux tower observations and benchmarking against widely used $T_s$ products. The proposed methodology could directly address the scaling challenges induced by spatial heterogeneity and uneven data distribution. The generated products would provide a robust foundation for high-resolution environmental modeling, precision agriculture and climate impact assessments.

## 2. Materials and methods

### 2.1 In-situ $T_s$ observations

In this study, in-situ $T_s$ observations was measured at six depths: at the surface (0 m), and at subsurface levels of 0.05, 0.10, 0.15, 0.20, and 0.40 meters. Data were collected through the national weather station network operated by the China Meteorological Administration (CMA), in accordance with standardized measurement protocols. At each site, $T_s$ was recorded every 10 minutes and automatically uploaded to a central server. Daily mean values at each depth were calculated from these high-frequency records. We then assessed data completeness for the period 2010–2020 and

excluded stations with more than 20% missing daily records at any depth. After quality control, 2,093 stations were retained for model development.

The observation network spans a wide range of climatic zones—from cold and temperate to subtropical and tropical, and includes diverse land-use and ecosystem types, such as forests, grasslands, croplands, and barren lands. However, the spatial distribution of stations is notably uneven. High station density is observed in northeastern China, the central and eastern plains, and the southern hilly regions, whereas station coverage is sparse in the arid and semi-arid regions of northwestern China and on the QTP. The spatial distribution of in-situ observation sites is shown in Figure 1, and details of the dataset partitioning strategy are provided in Section 2.3.3.

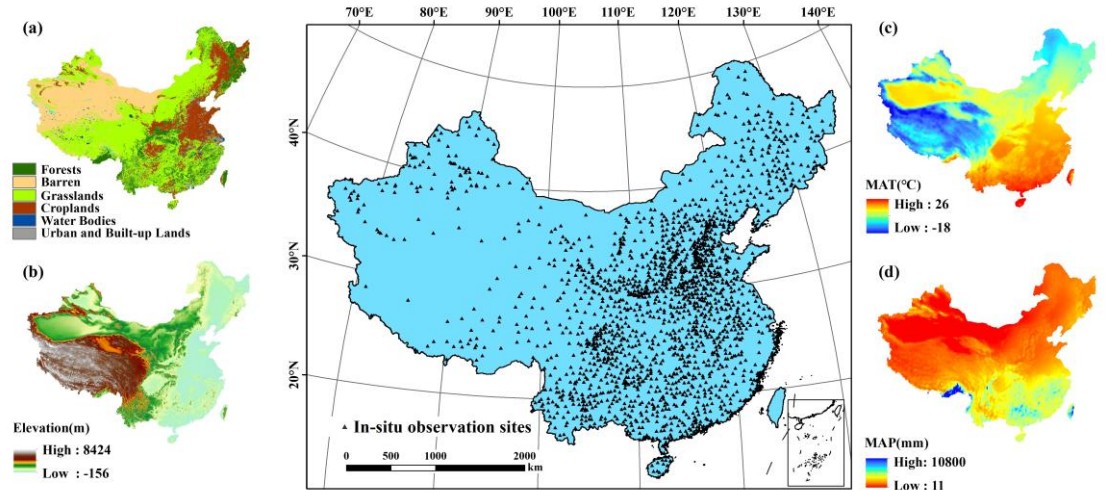

**Figure 1.** Spatial distribution of in-situ $T_s$ sites at different depths across China and the corresponding environmental variables. This figure presents the spatial distribution of 2,093 in-situ $T_s$ sites across China. The environmental variables corresponding to these sites include (a) land cover types (forests, barren land, grasslands, croplands, water bodies, and urban areas), (b) elevation (ranging from -156 m to 8424 m), (c) mean annual temperature (MAT, ranging from -18°C to 26°C), and (d) mean annual precipitation (MAP, ranging from 11 mm to 10,800 mm).

2.2 Predictor variables

To construct a robust multi-layer $T_s$ estimation model, we selected a comprehensive suite of predictor variables, integrating remote sensing products, meteorological factors, and auxiliary environmental data. Meteorological variables, especially air temperature and precipitation, have been consistently recognized in previous studies as primary determinants of $T_s$ variability (Bond-Lamberty et al., 2005;

Nahvi et al., 2016). Among these, air temperature has been widely regarded as the most influential variable due to its strong linear relationship with $T_s$ (Khosravi et al., 2023).

In addition, both net solar radiation and downward longwave radiation (LWD) were considered. Net solar radiation directly represents the shortwave energy absorbed by the land surface and serves as the primary driver of the daytime surface energy

budget, whereas LWD plays a particularly important role under nighttime and winter conditions by regulating surface heat loss through the longwave radiation balance. Together, they jointly control the surface energy balance and directly drive the spatiotemporal dynamics of $T_s$ (Peng et al., 2016).

Thermal infrared remote sensing data also exhibit a high correlation with near-

surface $T_s$. Integrating thermal remote sensing products and energy balance-based models offers an effective means of estimating $T_s$ with high spatial and temporal continuity. This strategy has been validated by numerous studies (Huang et al., 2020; Xu et al., 2023). Surface land cover further modulates $T_s$ by altering surface albedo, regulating evapotranspiration (ET), and influencing energy partitioning processes.

Accordingly, the enhanced vegetation index (EVI), derived from satellite observations, was incorporated as a proxy for vegetation density and type (Bright et al., 2017; Li et al., 2024b). To capture the influence of underlying surface characteristics on $T_s$, topographic variables such as elevation and slope were included, along with soil texture data across various depths. These features collectively reflect the heterogeneous

physical and thermal properties of the soil, contributing to spatial variations in heat conduction and storage capacity. A full list of the predictor variables used in the model is summarized in Table 1.

**Table 1.** Details of the predictor variables for training the model.

| Type | Data | Variable | Spatial resolution | Temporal resolution | Reference |
|---|---|---|---|---|---|
| Remotely sensed product | MOD09GA | EVI | 500 m×500 m | Daily | Huete et al., 2002 |
| | MOD11A1 | LST_ Day | 1 km×1 km | Daily | |
| | MOD11A1 | LST_ Night | 1 km×1 km | Daily | |
| Climate data | ERA5-Land | Temperature_2m surface_net_solar_r adiation_sum surface_thermal_ra diation_downwards _sum Precipitation | 9 km×9 km | Daily | Muñoz-Sabater et al., 2021 |
| Supplementary data | USGS_STRM | Elevation | 30 m | | |
| | | Slope | 30 m | | |
| | Soil Texture | Sand, Silt, Clay Depth: 0-5, 5-15, 15-30, 30-60cm | 250 m×250 m | | Liu et al., 2022 |
| | In-situ measurements | Soil temperature at 0, 5, 10, 15, 20, and 40 cm | - | Daily | |

2.2.1 Remote sensing data

The MOD11A1 LST product, at a daily time-step and a spatial resolution of 1 km, was utilized. It includes both daytime ($LST_{day}$) and nighttime ($LST_{night}$) temperatures at 10:30 AM and 10:30 PM, respectively, along with quality assessment information (Wan and Dozier, 1996). To enhance the estimation of daily mean $T_s$, the average of $LST_{day}$ and $LST_{night}$ values was calculated and used in the analysis.

EVI from 2010 to 2020 were selected as predictor of $T_s$. The MODIS Surface Reflectance Product (MOD09GA), derived from MODIS Level-1B data, provides daily surface reflectance of seven bands at 500 m × 500 m resolution. The EVI is defined by Huete et al., (2002), and the retrieval equation is as follows:

$$EVI = G \times \frac{\left(\rho_{SR\_b1} - \rho_{SR\_b2}\right)}{\left(\rho_{SR\_b1} + C_1 \times \rho_{SR\_b2} - C_2 \times \rho_{SR\_b3} + L\right)} \tag{1}$$

where G = 2.5, $C_1$ = 6, $C_2$ = 7.5, L = 1. The remote sensing reflectance variables

*SR_b1*(620-670nm), *SR_b2* (841-876nm) and *SR_b3* (459-479 nm) of MOD09GA data represents red, near-infrared and blue bands. The coefficients 2.5 and 1 represent the gain and canopy background, respectively (Huete et al., 2002). The atmospheric influence on the red band is corrected using the blue band and the coefficients 6 and 7.5, respectively.

Subsequently, cloud contamination caused partial spatial absences in the daily LST and EVI. To address this issue, we applied a temporal and spatial linear interpolation algorithm, which utilizes time-series data from adjacent days and spatial information from neighboring pixels to fill the current missing values, thereby generating a time-continuous and spatially complete image series. This approach follows the methods described in Chen et al., (2017) and Cao et al., (2018), with modifications to better suit our dataset. Then, the Savitzky-Golay (S-G) filter was used to smooth the interpolated data, resulting in continuous surface temperature and vegetation index data with high temporal and spatial resolution (Kong et al., 2019; Chen et al., 2021b). All data preprocessing, including image filtering and interpolation, was conducted within the Google Earth Engine (GEE) platform.

2.2.2 Climate data

The ERA5-Land is the fifth-generation reanalysis dataset produced by the European Centre for Medium-Range Weather Forecasts (ECMWF). It assimilates multi-source data, including weather station measurements, numerical weather predictions, and satellite observations, into dynamic models to generate reanalysis data (Muñoz-Sabater et al., 2021). It provides high-quality environmental variables related to water and energy fluxes between the land surface and atmosphere, with continuous coverage from 1981 to the present. ERA5-Land offers a spatial resolution of 0.1° (~9 km at the equator) and an hourly temporal resolution, making it well-suited for modeling near-surface processes. In this study, we extracted daily mean values of key climate variables, including 2-meter air temperature (Temperature_2m), surface solar radiation and total precipitation from the ERA5-Land Daily dataset. All variables were accessed and processed using the GEE platform.

### 2.2.3 Auxiliary data

Topographic and soil-related variables were incorporated as auxiliary predictors to improve the accuracy of $T_s$ estimation. Elevation and slope were derived from the Shuttle Radar Topography Mission (SRTM) digital elevation model (Farr et al., 2007), specifically using the Version 3 (SRTM Plus) product with a spatial resolution of 1 arc second (~30 m). Soil texture plays a critical role in determining $T_s$ through its influence on thermal conductivity, which is affected by physical properties such as particle size distribution, porosity, bulk density, and moisture retention capacity. In this study, we represented soil texture using the relative proportions of clay (fine), silt (medium), and sand (coarse) particles. To capture vertical variability in soil properties, we employed the China Soil Information Grid dataset developed by Liu et al. (2022), which provides gridded estimates of soil composition at four depth intervals: 0~5 cm, 5~15 cm, 15~30 cm, and 30~60 cm. The dataset offers a spatial resolution of 1 km and is suitable for high-resolution, profile-based soil modeling.

### 2.3 Methods

The spatial adaptive modeling framework consists of three modules as shown in Fig. 2. Module I is for data collection and preprocessing, which mainly involves in-situ observations, remote sensing, meteorological and supplementary data. Module II is spatial adaptive modeling, which mainly includes the construction of rotated quadtrees and local modeling based on XGBoost. Finally, module III is the layer-to-layer reconstruction of daily 1 km resolution multi-layer (0, 5, 10, 15, 20, and 40 cm) $T_s$ datasets in China from 2010 to 2020.

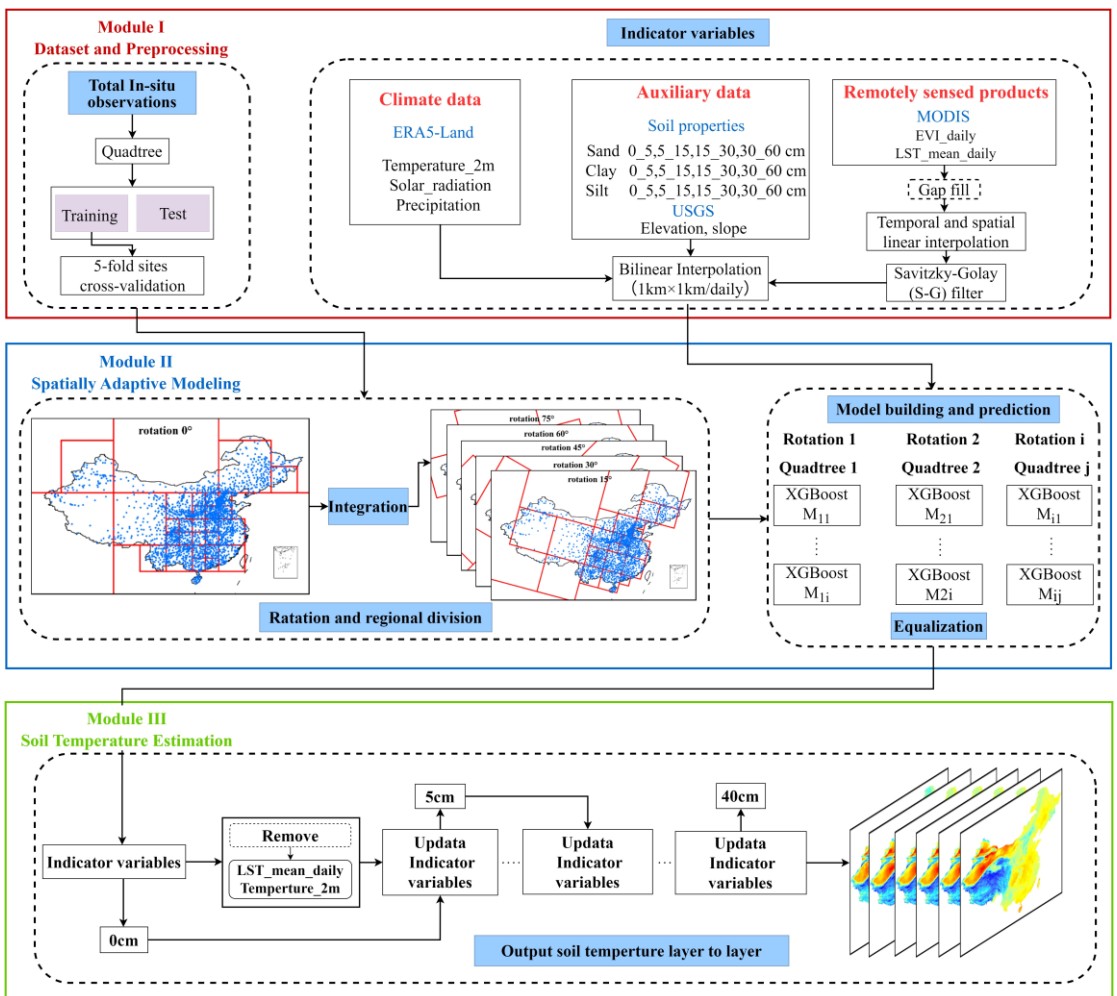

**Figure 2.** Workflow of the proposed method to obtain multi-layer $T_s$ over the China.

### 2.3.1. Feature selection

Multicollinearity among multiple source variables may affect the robustness of the models. Therefore, we rigorously evaluated the multicollinearity among the independent variables using the variance inflation factor (VIF) before modeling to remove highly correlated variables. The VIF is a diagnostic statistic used to quantify the degree of multicollinearity by measuring how much the variance of a regression coefficient is inflated due to correlations with other predictors (Akinwande et al., 2015). It is calculated as:

$$VIF_i = \frac{1}{1 - R_i^2} \tag{2}$$

where $R_i^2$ is the coefficient of determination obtained by regressing the $i$-th predictor against all other predictors. Variables with VIF exceeding 10 are generally considered

severely multicollinear and should be removed.

Based on the VIF analysis, we applied the following adjustments to the predictor set. Accordingly, some variables were excluded due to severe multicollinearity or redundancy. Specifically, sand, silt, and clay are compositional variables whose proportions sum to 100%, leading to perfect collinearity. To reduce redundancy, we removed silt while retaining sand and clay. In addition, LWD was found to be highly correlated with net solar radiation at the daily mean scale (Fig. S1) and was therefore excluded from the final modeling.

Although the daily mean LST (LST_mean) and air temperature exhibit high collinearity (VIF > 10; Fig. S2), we chose to retain both variables because they represent different thermal information. LST_mean captures high-resolution surface radiative temperature signals, whereas air temperature reflects broader-scale atmospheric thermal conditions. In ecosystems with complex canopy structures, such as forests, the canopy can alter radiative transfer processes and cause LST to deviate from the true subsurface thermal environment(Liu et al., 2025). Therefore, the two variables provide complementary thermal information that helps better characterize soil thermal dynamics. In addition, we compared the model performance under different combinations of predictor variables (Fig. S3 and Fig. S4). The results show that the combination of air temperature + LST + other predictors achieved the best modeling accuracy at the surface soil layers. Therefore, retaining both air temperature and LST in the final model is reasonable and necessary.

### 2.3.2. Spatial adaptive partition of site measurements

We applied the Local Bivariate Moran's I analysis to assess the local spatial relationship between surface $T_s$ (GST_Avg) and elevation as an illustrative example (Fig. S5). The results reveal significant spatial variations in their local association ($p < 0.05$), indicating pronounced spatial non-stationarity in the $T_s$–elevation relationship. These findings justify the need for a spatially adaptive modeling strategy capable of capturing localized heterogeneity.

A quadtree is a hierarchical spatial data structure that recursively subdivides a two-dimensional space into four quadrants, enabling efficient spatial indexing and localized data organization. In this study, we adopted a bottom-up, rotated quadtree-based spatial partitioning strategy that adaptively generates finer grids in regions with dense samples and coarser grids in sparse regions. Compared to global modeling or static grid partitioning, this adaptive approach offers improved regional modeling fidelity while significantly enhancing computational efficiency. The procedure consists of the following steps:

(1) Initialization of Minimum Units

The entire spatial domain was first divided into uniform, minimum-sized units (leaf nodes), each representing a fundamental spatial element. These units may contain zero or more in-situ observations. This initial step provides the base resolution for subsequent hierarchical construction. The structure and principle of quadtree spatial indexing are illustrated in Fig. S6.

(2) Hierarchical Merging

Starting from the leaf nodes, groups of four adjacent quadrants were recursively merged into parent nodes if each contained fewer than 30 observation sites (threshold selection detailed in Fig. S7). The merging process continued upward until no further groups met the threshold. This approach ensures that each node has sufficient sample size while achieving spatially adaptive partitioning across the study area. Each subregion is then assigned a localized $T_s$ prediction model.

(3) Rotation at different angles

To reduce potential edge effects introduced by static grid boundaries, we implemented a rotated quadtree partitioning strategy. The quadtree structure was rotated at six angles (0°, 15°, 30°, 45°, 60°, and 75°), producing distinct sets of spatial partitions for each orientation (Fig. 3). Independent models were trained for each rotated configuration, and the final $T_s$ estimates were obtained by averaging the outputs from all six models. This rotation-based ensemble method improves spatial smoothness and minimizes discontinuities at partition boundaries.

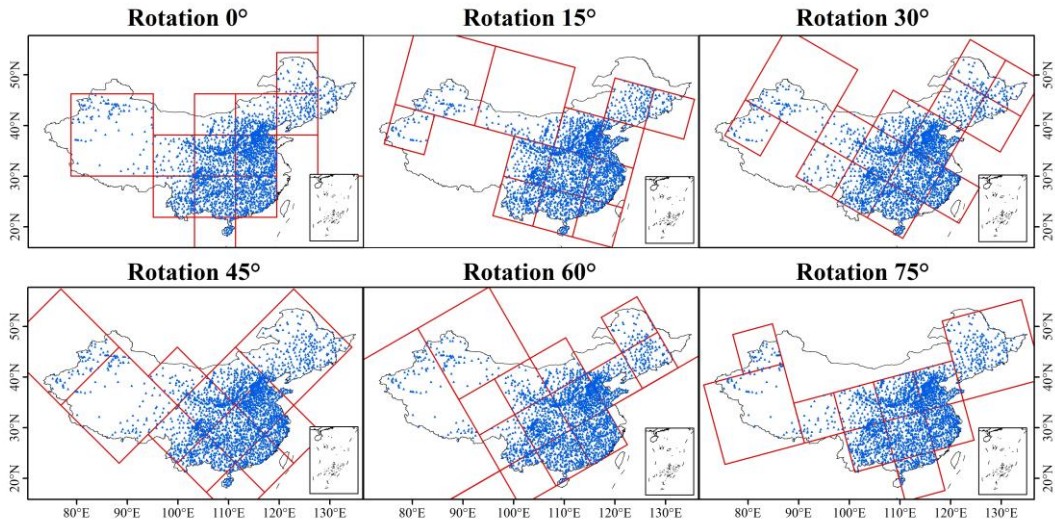

**Figure 3.** Multi-angle adaptive quadtree partitioning of site observations (0°, 15°, 30°, 45°, 60°, 75°)

2.3.3. Machine learning algorithm

We adopted the XGBoost (Extreme Gradient Boosting) algorithm as the core regression model for $T_s$ estimation due to its strong predictive performance, computational efficiency, and scalability across large environmental datasets. XGBoost constructs an ensemble of regression trees in a stage-wise boosting process, where each successive tree is trained to minimize the residuals of the previous iteration, thereby producing a robust and optimized model (Chen and Guestrin, 2016). One of the key strengths of XGBoost is its ability to handle heterogeneous and high-dimensional predictor sets, which are common in geoscience applications involving complex terrain, land cover variability, and climatic gradients. Recent studies have demonstrated its effectiveness in similar domains, including land surface temperature reconstruction (Li et al., 2024), multi-layer soil moisture estimation (Karthikeyan and Mishra, 2021), drought event attribution (Wang et al., 2025a), and crop yield prediction (Li et al., 2023b). Given these proven strengths and the spatially nonstationary characteristics of $T_s$ in our study area, XGBoost was selected to train localized prediction models within spatial subregions.

Significant spatial autocorrelation commonly exists among nearby $T_s$ observation sites. To prevent potential data leakage caused by randomly splitting the training and testing subsets, we conducted the partitioning at the station level and constructed a

buffer zone around the selected test station. All other stations located within this buffer were removed, and only stations outside the buffer were retained as the training set. This strategy effectively ensures that samples within the same sub-grid do not appear simultaneously in both the training and testing subsets due to spatial autocorrelation, thereby allowing a more robust and unbiased assessment of the model's generalization performance.

Specifically, considering the availability of sufficient training samples, one station was randomly selected as the test sample within each sub-grid. A 500 km buffer was subsequently created around the test station, with the radius determined based on the effective distance for reducing spatial autocorrelation among stations as shown in Appendix Figure S8. All stations within the buffer were excluded, and only those outside the buffer were used for model training. Subsequently, five-fold cross-validation was performed at the station level, and GridSearchCV was used to optimize three key hyperparameters: the number of trees (n_estimators), maximum tree depth (max_depth), and learning rate (learning_rate). The search ranges for these parameters are provided in Appendix Table S1. The optimal hyperparameter combination was identified by minimizing the mean validation error. Finally, the model was retrained on the full training subset using the optimized parameters and evaluated on the spatially independent test sample to rigorously assess its generalization capability.

A layer-wise prediction strategy was adopted to estimate $T_s$ along the soil profile. For the surface layer (0 cm), predictors included air temperature and daily mean LST. For subsurface layers, these two variables were replaced by the $T_s$ estimate from the immediately preceding layer, enabling the model to capture vertical heat conduction processes and thereby improving the continuity and physical consistency of layer-wise $T_s$ estimation.

2.3.4. Model evaluation metrics

The modeling performance and quality of the predicted $T_s$ were evaluated in terms of RMSE, Mean Absolute Error (MAE), $R^2$, and Bias. RMSE and MAE were used to

385 assess the ability to estimate volatility and fluctuation amplitude, respectively. $R^2$ represented the percentage of variance explained by the ML models. Bias was used to determine whether the estimations were overestimated or underestimated. These metrics were computed as follows:

$$RMSE = \sqrt{\frac{\sum_{i=1}^{N}[(x_i - \overline{X}) - (y_i - \overline{Y})]^2}{N}} \tag{3}$$

$$MAE = \frac{\sum_{i=1}^{N}|x_i - y_i|}{N} \tag{4}$$

$$Bias = \frac{1}{N}\sum_{i=1}^{N}(x_i - y_i) \tag{5}$$

$$R^2 = 1 - \frac{\sum_{i=1}^{N}(y_i - x_i)^2}{N\sum_{i=1}^{N}(y_i - \overline{Y})^2} \tag{6}$$

where $y_i$ and $x_i$ denoted the in-situ $T_s$ and estimated $T_s$ for all the stations and periods, respectively. $\overline{Y}$ and $\overline{X}$ represented the mean values of the in-situ $T_s$ and estimated $T_s$, respectively.

**3. Results**

3.1 Model performance across sites

Figure 4 shows the accuracy of the models constructed at different depths using various grid configurations and rotation angles for both the training and test sets. The grouped box plots indicate that the median $R^2$ values range from 0.92 to 0.98 and the

405 median RMSE values range from 1.6 to 2.4 K across depths. Both training and test results exhibit consistently high accuracy, with no clear indication of overfitting. A vertical comparison shows that model performance at 0 cm and 40 cm is slightly weaker than that at intermediate depths.

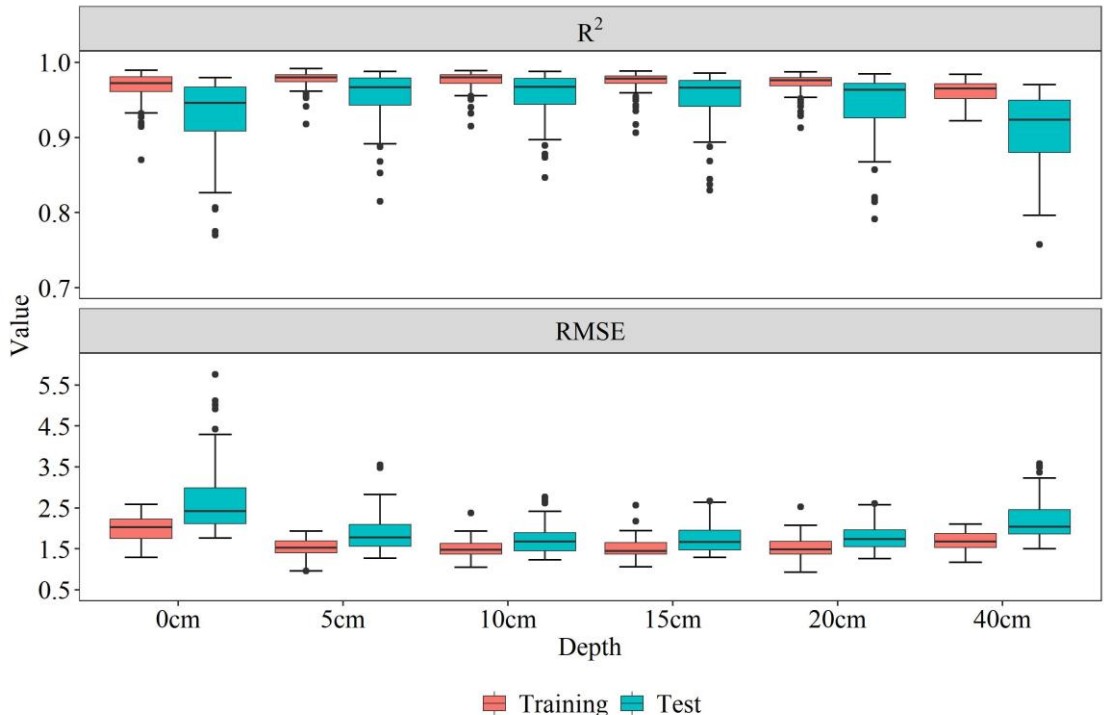

**Figure 4.** Model performance for training and test sets across different depths.

To further enhance the independence of the evaluation, we validated the final dataset using daily $T_s$ observations from 18 flux tower sites in the ChinaFLUX network. For consistency across depths, only measurements at 0, 5, 10, 15, 20, and 40 cm were retained. Metadata for these sites is summarized in Table S2, and the corresponding validation results are presented in Figure 5. The results show that the dataset maintains high accuracy at independent sites ($R^2 = 0.78\sim0.87$; RMSE = 3.89~5.14 K), further demonstrating the robustness of our approach. Overall, the combined evidence from the test set and flux tower validation confirms that the proposed spatially adaptive model exhibits strong predictive performance and spatial generalization capability. In Figure S9, we further validated the spatial consistency between the flux tower sites and the estimated annual mean $T_s$ at different depths. Although the validation results demonstrated high accuracy overall ($R^2 = 0.7\sim0.82$; RMSE = 2.93~3.58 K), a systematic positive bias of approximately +2 to +3 K was observed across all depths.

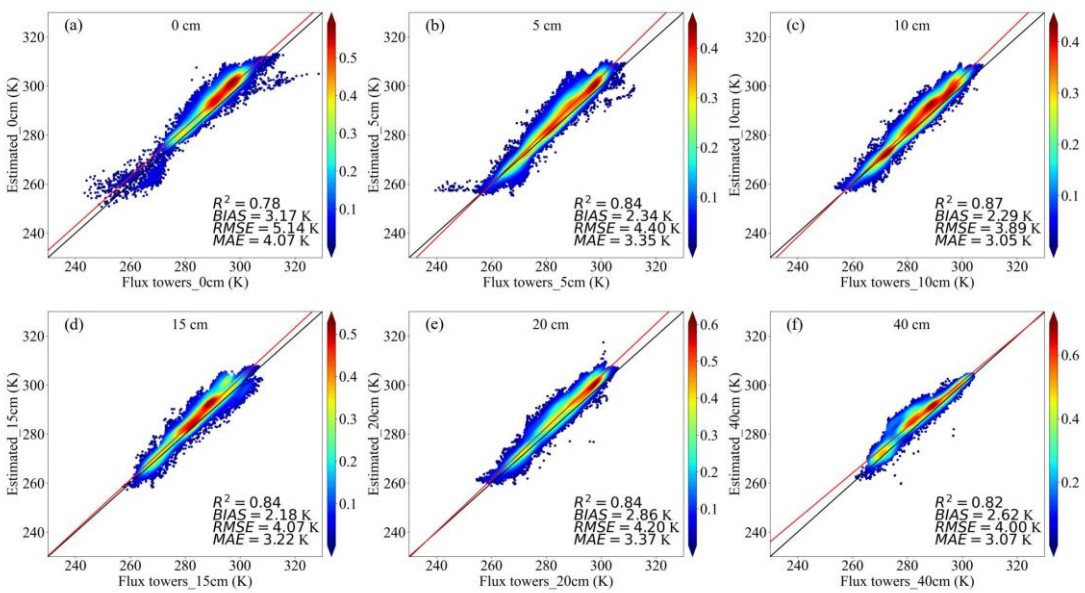

**Figure 5.** Density scatter plots comparing estimated daily $T_s$ with flux tower observations at different depths

We also calculated R² and RMSE values for all depths at each station to compare the model performance. The results indicate that R² ranges from 0.70 to 1.00, suggesting generally good performance at the station level. As shown in Figure 6, most stations achieve R² values above 0.85. Regions with higher prediction accuracy are primarily distributed across northwest, northeast, and central China, while larger errors are concentrated in the Yunnan–Guizhou Plateau (YGP) and the sparsely monitored QTP. The histogram in Figure S10 further shows that RMSE values for all depths fall between 0.5 and 3 K, indicating overall good predictive performance. Notably, prediction errors are highest at 0 cm, decrease substantially at 5–20 cm, and increase slightly again at 40 cm. Figure S11 shows the comparison between the estimated and observed annual mean $T_s$ for the test dataset at six different depths (0~40 cm). The R² ranges from 0.94 to 0.97. The RMSE values range from 0.74 to 1.4 K, and the bias is minimal. The results suggest that the model is able to effectively capture the spatial patterns of $T_s$ across different depths and locations.

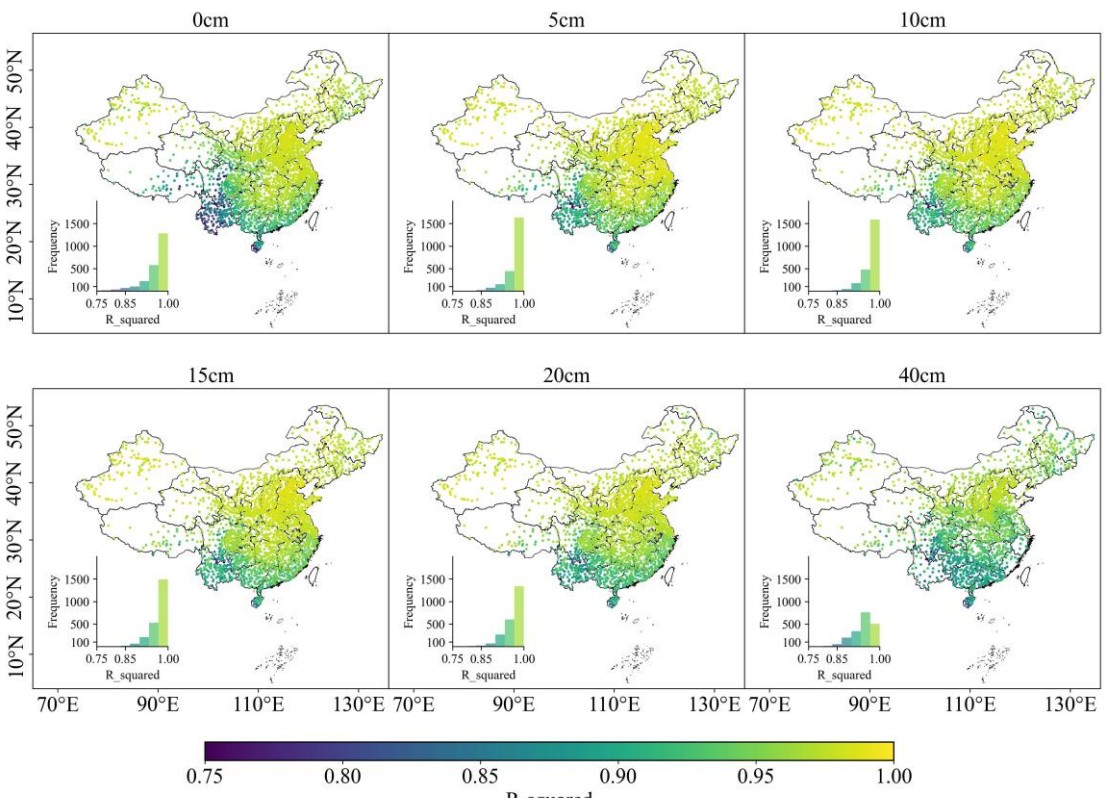

**Figure 6.** Goodness of R² across China estimated during the model testing phase. Performance metrics are calculated between predicted_$T_s$ and in-situ $T_s$ data sets.

3.2 Evaluation across land cover types and seasons

Figure 7 shows grouped box plots of the prediction performance of $T_s$ across different land cover types (barren land, cropland, forest, and grassland) at six depths (0, 5, 10, 15, 20, and 40 cm). The evaluation metrics include R² and RMSE. The median R² values across land cover types and depths range from 0.94 to 0.98, consistently exceeding 0.94 (red dashed line), indicating overall high prediction accuracy. Among land cover types, barren land exhibits the highest R² values, followed by cropland, while forest and grassland show slightly lower performance. The median RMSE values generally range from 1.1 to 1.8 K. Barren land shows higher RMSE compared with other land cover types, whereas cropland, forest, and grassland maintain lower and more stable RMSE. Across depths, RMSE is highest at the surface layer (0 cm), decreases steadily with increasing depth, and shows a slight increase at 40 cm.

Furthermore, seasonal variations in prediction accuracy are shown in Fig. 8. The median R² values across depths range from 0.48 to 0.98, with higher values in spring (green) and autumn (pink) and lower values in summer (orange) and winter (blue),

particularly at 20~40 cm depth. The median RMSE values range from approximately 1.3 to 2.2 K, being lower in spring and autumn and higher in summer and winter, with the largest median error observed at 40 cm depth in winter. With increasing depth, the median errors decrease from the surface (0 cm) to 5~10 cm, and then gradually

accumulate from 15 to 40 cm.

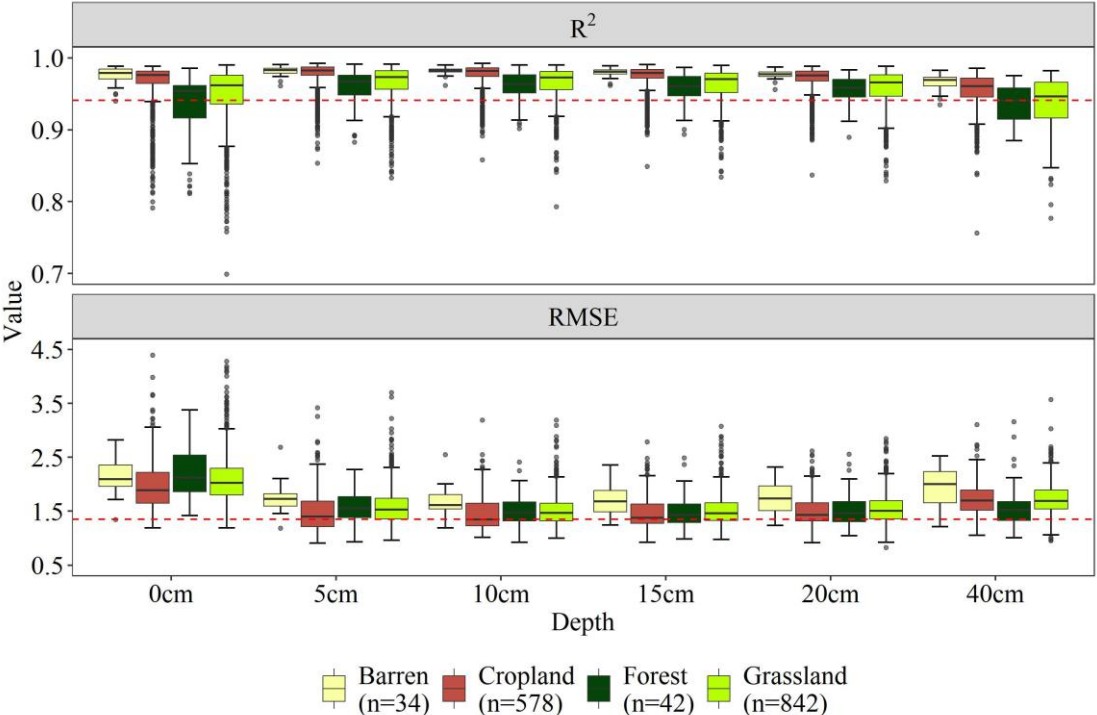

**Figure 7.** Evaluation of predicted $T_s$ at different depths (i.e., 0, 5, 10, 15, 20, 40cm) across various land use types (i.e., Forest, Grassland, Cropland, Barren)

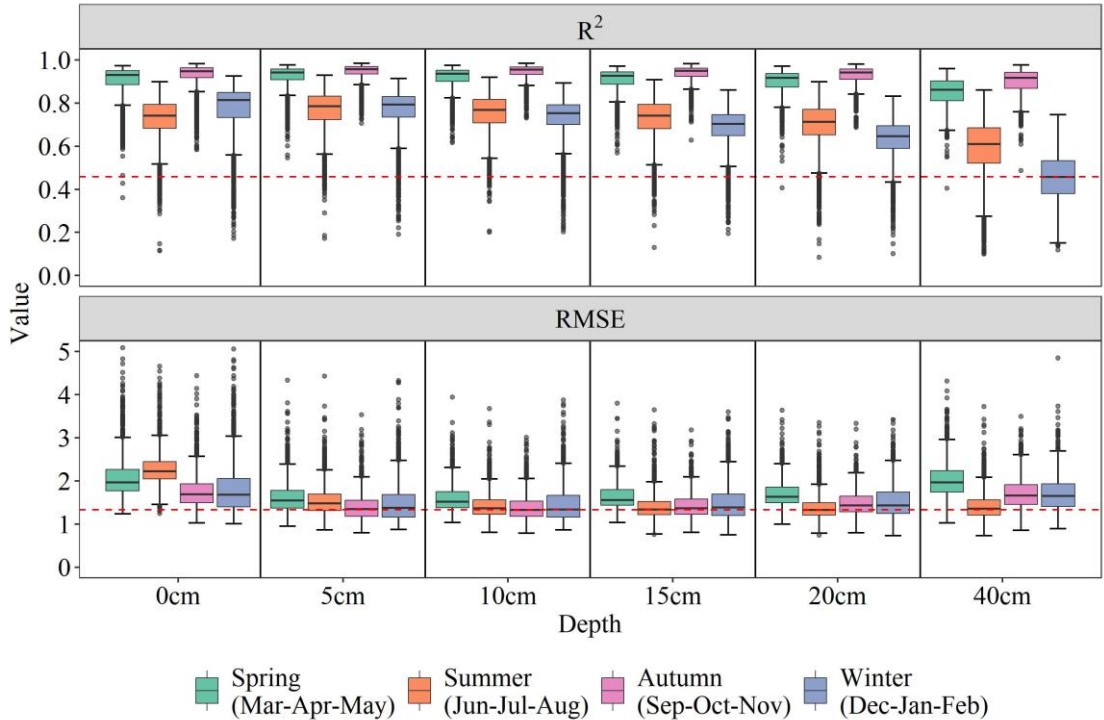

**Figure 8.** Evaluation of the predicted_$T_s$ in different depth (ie.0,5,10,15,20,40 cm) at sites with four seasons (i.e., spring, summer, autumn, winter). Winter is defined as December, January, and February; spring as March, April, and May; summer as June, July, and August; and autumn as September, October, and November.

3.3 Comparison with other products

Figure 9 presents a comparison of the $T_s$ products at the 0 cm depth with the ERA5-Land and GLDAS 2.1 reanalysis datasets, including both national-scale patterns (Fig. 9a–c) and zoomed-in regional details (Fig. 9d–f). Compared with the two reanalysis products, our generated $T_s$ dataset exhibits substantially finer spatial resolution, enabling a clearer representation of localized spatial heterogeneity. As illustrated in the zoomed-in panels of Figure 9, our $T_s$ product accurately captures terrain- and elevation-driven temperature gradients in regions with strong topographic variability, such as the transition zone from the Sichuan Basin to the margins of the QTP. In contrast, the coarse spatial resolution of ERA5-Land and GLDAS 2.1 tends to smooth out these fine-scale topographic effects, resulting in a loss of spatial detail.

The scatter density plots in Fig. S12 further demonstrate that the $T_s$ estimates from our model achieve significantly higher site-level accuracy than ERA5-Land and GLDAS 2.1. Specifically, at depths of 0, 10, and 40 cm, the $R^2$ values for our dataset range from 0.94 to 0.97, whereas the corresponding values are 0.83~0.89 for ERA5-

Land and 0.83~0.87 for GLDAS 2.1. These results indicate that our high-resolution $T_s$ product not only captures localized heterogeneity but also faithfully represents terrain-driven temperature gradients, which are often obscured in coarse-resolution reanalysis products. In summary, the proposed spatially adaptive modeling framework provides a more detailed and realistic representation of $T_s$ spatial patterns, particularly in

topographically complex regions, and significantly enhances the accuracy and applicability of regional-scale $T_s$ modeling.

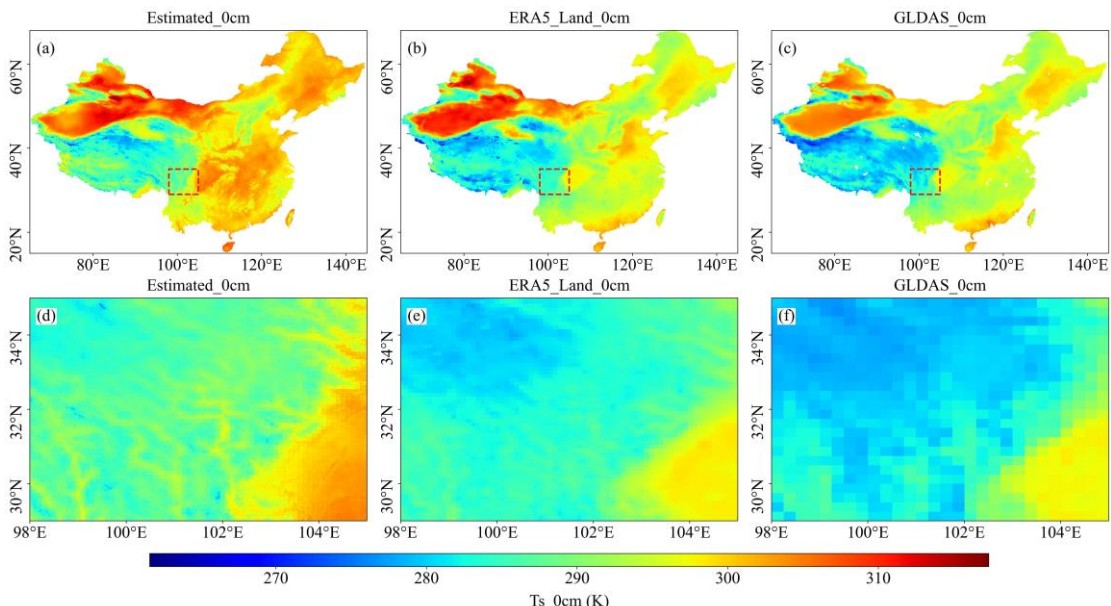

**Figure 9.** Comparison of different $T_s$ products (e.g., 0 cm)

3.4 Spatial and temporal patterns of $T_s$ at varied depths across China

To examine seasonal and vertical variations in the spatial distribution of $T_s$, we selected two contrasting dates: January 1, 2020 (winter) and July 1, 2020 (summer). Figure 10 a–f illustrates the spatial distribution and corresponding histograms of $T_s$ at different depths (0 cm, 5 cm, 10 cm, 15 cm, 20 cm, 40 cm) across China on January 1, 2020. The results show that $T_s$ in northern China (particularly in the northeast, northwest,

and the QTP) is generally lower in January, exhibiting distinct cold zones. In contrast, southern areas exhibit higher $T_s$ values, forming a gradual north-to-south temperature gradient. Moreover, deeper soil layers (e.g., 40 cm) exhibit higher temperatures than surface layers (0 cm), especially in northeastern China and the QTP, reflecting the insulating effect of deeper soils during winter.

Figure 10a1–f1 illustrates the spatial distribution and histograms of $T_s$ on July 1,

2020. Compared to January, a significant increase in $T_s$ is observed across China in July, with widespread high-temperature zones in the eastern and southern regions. The increase is particularly pronounced in northern areas, while changes in the south are relatively moderate. In contrast to winter conditions, $T_s$ decreases with increasing soil depth during summer, with surface temperatures (0 cm) exceeding those at 40 cm, indicating the downward heat conduction from the surface. Overall, Comparative analysis of Fig. 10a–f and Fig. 10a1–f1 elucidates both seasonal variation and vertical patterns of $T_s$: deeper layers (5~40 cm) are warmer than the surface (0 cm) during winter, whereas the surface is warmer in summer. The histogram further illustrates the variation in $T_s$ distribution across different depths. The results indicate that temperature fluctuations in deeper layers are significantly smaller than those near the surface, reflecting greater thermal stability in the subsurface. These patterns reflect the combined influences of geographic location, topography, and climatic conditions on $T_s$ spatial distribution and vertical dynamics, offering valuable insights into soil thermal behavior.

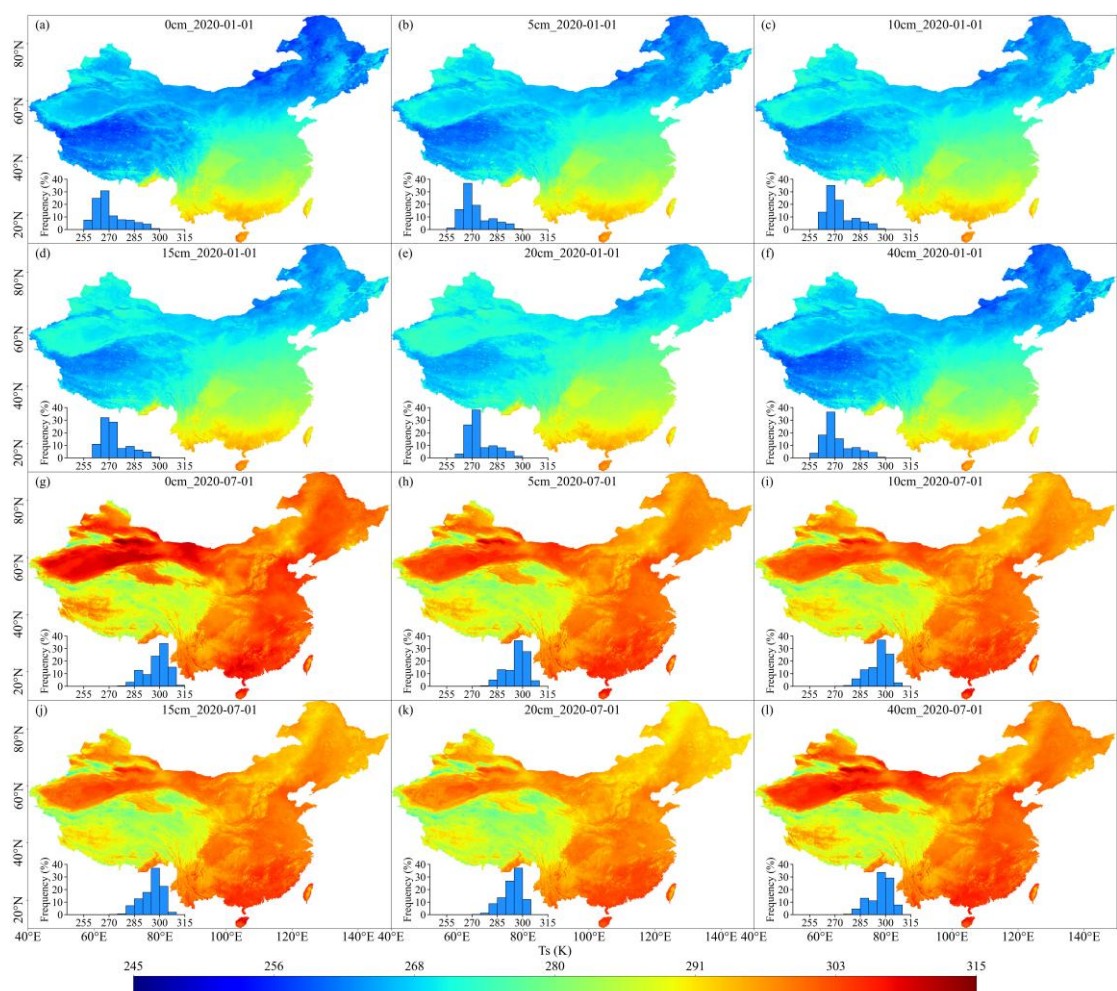

**Figure 10.** Spatial patterns and histograms of Estimated $T_s$ at different depths (0, 5, 10, 15, 20, and 40 cm)

To further assess the temporal performance of $T_s$ estimation, Fig. 11 presents the time series of estimated $T_s$ alongside in-situ measurements at four randomly selected stations (e.g., Station 56748, 99.18°E, 25.12°N) from January 2018 to January 2020. The figure displays $T_s$ at two depths (0 cm and 40 cm), including estimated $T_s$ (Estimated_0cm, Estimated_40cm), in-situ $T_s$ (In-situ_0cm, In-situ_40cm), daily mean land surface temperature (Daily_mean_LST), and 2-meter air temperature (Temperature_2m). The air temperature shows distinct seasonal cycles, while $T_s$ exhibits smoother temporal variations. In general, $T_s$ reaches its peak during summer and its minimum in winter, though its temporal dynamics vary with soil depth. Specifically, $T_s$ at 0 cm responds rapidly to air temperature changes and exhibits larger amplitude variations, while $T_s$ at 40 cm shows slower responses and a noticeable lag, reflecting the damping effect of vertical heat conduction. Site-level accuracy was

evaluated using RMSE, which ranged from 1.24 K to 2.05 K across both depths, indicating strong agreement between predicted and observed values. Overall, the time series analysis confirms the robustness and reliability of the model in estimating $T_s$ across varying depths, offering valuable insights into regional soil thermal dynamics.

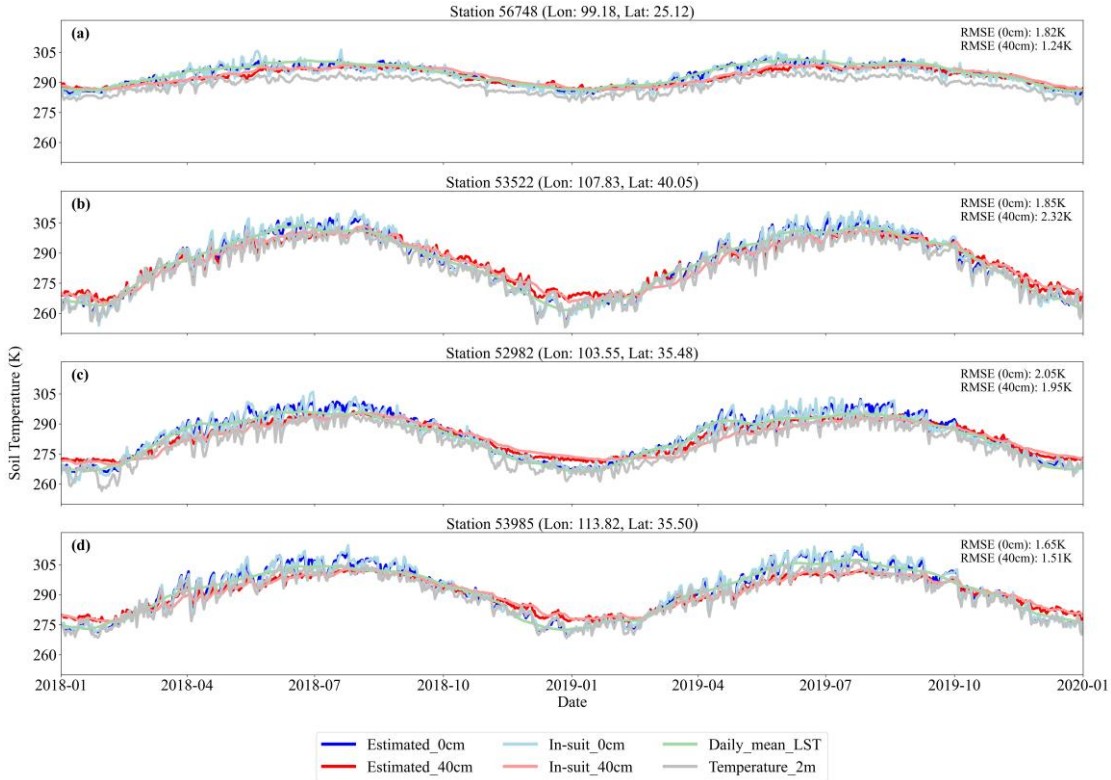

**Figure 11.** Time series of the Estimated_0cm, Estimated_40cm, Daily_mean_LST, and Temperature_2m at four sites from different regions between 2018-2019.

## 4. Discussion

### 4.1 The advantages of the spatially adaptive model

Previous studies have explored various approaches for constructing $T_s$ datasets. For instance, Wang et al., (2023) created a daily multi-layer $T_s$ dataset for China (1980-2010) at 0.25° resolution, employing interpolation techniques including the thin-plain spline and the angular distance weight interpolation methods with over 2,000 in-situ observations. A persistent challenge in building national-scale $T_s$ datasets, however, lies in the highly uneven spatial distribution of observation stations—densely clustered in eastern lowlands while remaining sparse in western and high-altitude regions. Global modeling approaches, which train a single unified function across the entire domain, are inherently limited in capturing the nonlinear and non-stationary relationships

between $T_s$ and its predictors in such heterogeneous landscapes. Specifically, in sparsely sampled regions, global models lack sufficient data to learn effectively, resulting in low prediction accuracy. In contrast, in densely sampled areas, the model tends to overfit, and the training process becomes disproportionately influenced by those regions. This imbalance introduces systematic biases and limits model generalizability.

Reanalysis datasets, which synergize data assimilation systems with numerical weather prediction and land surface modeling frameworks, provide valuable representations of land-atmosphere interactions and subsurface heat transfer processes. These products are particularly advantageous for large-scale climate simulations and long-term environmental assessments. Yang and Zhang (2018) assessed the $T_s$ accuracy

of four reanalysis datasets (ERA-Interim/Land, MERRA-2, CFSR, and GLDAS-2.0) in China using in-situ monthly mean $T_s$ observations. The results showed that all reanalysis datasets consistently underestimated $T_s$ across the country. More recently, the ERA5-Land and GLDAS 2.1 $T_s$ dataset offers high temporal resolution (hourly/3-hour), but it is limited by a spatial resolution of 0.1 or 0.25 degrees. Beyond reanalysis datasets,

some efforts have focused on constructing empirical $T_s$ products using ML approaches. For example, the Global Soil Bioclimatic Variables dataset (Lembrechts et al., 2022), derived from Random Forest modeling with 8,519 global sensors, provides only long-term climatological means, rather than high-resolution daily estimates.

In contrast, the methodological framework proposed in this study addresses both

accuracy and resolution limitations. The spatially adaptive modeling strategy offers significant advantages over traditional interpolation and globally trained ML models. Its core strength lies in localized modeling, which accounts for regional variability in topography, soil properties, and climate conditions. As shown in Fig. S13, the rotated quadtree strategy partitions space at six orientations (0°~75°), enabling a more nuanced

representation of spatial heterogeneity. By averaging predictions across these rotated configurations, the method reduces boundary artifacts often associated with static grids, resulting in smoother and more continuous spatial outputs. We also quantified the variability of prediction results at the same site using grids generated from different

rotation angles. The results in Fig. S14 show that the uncertainty at the 0 cm depth is higher compared to other depths, with the highest uncertainty concentrated in certain areas of the YGP and Sichuan Basin.

Moreover, the fine spatial resolution (1 km) enables the model to resolve localized thermal patterns that are critical for understanding vegetation dynamics and soil biogeochemistry. We also assessed the contribution of satellite-derived LST to model performance. As shown in Figs. S3 and S4, incorporating LST as an input variable, relative to using only air temperature, significantly enhances overall modeling accuracy and improves performance across sites with different land cover types, with the most pronounced improvements observed in barren land areas. This highlights the importance of multi-source data fusion in boosting the performance of spatially adaptive models under data-scarce conditions. In summary, our spatially adaptive local modeling approach offers a more robust and scalable solution for large-scale $T_s$ estimation under heterogeneous station distributions and complex environmental conditions.

4.2 Potential applications of the $T_s$ product

The high-resolution, multi-layer $T_s$ datasets generated using the spatially adaptive estimation method fill a significant data gap in China, where comprehensive $T_s$ profile records are scarce. As a key biophysical variable, $T_s$ provides crucial insights into soil–atmosphere interactions that are not captured by air temperature alone. In agricultural systems, $T_s$ governs fundamental processes throughout the crop life cycle—from sowing and germination to growth and yield formation (Rahman et al., 2019). Multi-layer $T_s$ data can optimize accumulated temperature models, enhancing the precision of sowing decisions and supporting sustainable field management. Additionally, $T_s$ influences nutrient decomposition and water movement within soil profiles (Jebamalar et al., 2012), directly impacting soil fertility, moisture retention, and thus, the overall efficiency of agroecosystems.

Beyond agricultural applications, $T_s$ is increasingly recognized as a critical variable for assessing ecosystem responses to climate extremes. For instance, Fan et al.,

(2024) proposed the Soil Composite Drought Heatwave index to evaluate the severity of concurrent drought and heatwave events. However, their findings show that existing reanalysis datasets often underestimate these events compared to observational records, highlighting the need for more accurate, high-resolution $T_s$ data. In the context of intensifying global warming and extreme climate events, access to reliable $T_s$ datasets is essential for improving the monitoring and prediction of environmental stressors. These advancements are not only vital for understanding terrestrial ecosystem dynamics but also for strengthening climate resilience at both regional and national scales.

Moreover, $T_s$ plays a pivotal role in ecological and hydrological modeling, offering a more direct representation of surface processes than air temperature. It serves as a sensitive indicator of biogeochemical cycles and phenological changes (Lembrechts et al., 2022). For example, Liu et al., (2024) demonstrated that $T_s$ is a dominant driver of spring phenology in Chinese forests, making it a valuable input for climate–vegetation interaction models. In cold regions, $T_s$ governs soil freeze–thaw cycles, which are critical for hydrological processes such as runoff generation, groundwater recharge, and permafrost monitoring (Smith et al., 2022; Xu et al., 2022). Furthermore, $T_s$ is a key driver of soil respiration, influencing $CO_2$ fluxes and terrestrial carbon cycling (Lloyd and Taylor, 1994; Hursh et al., 2017). As such, the development of high-resolution $T_s$ products enables more accurate simulation of ecosystem carbon dynamics and regional carbon budgeting, thereby advancing our understanding of climate feedback mechanisms.

4.3 Limitations and future perspective

Despite the strong performance of our spatially adaptive $T_s$ estimation framework, several limitations warrant acknowledgment. As shown in Figure 6, model validation at station level reveals spatial heterogeneity in prediction accuracy, with relatively lower performance observed in the YGP and the QTP regions. On the one hand, as evidenced by Figure 9, our multi-source modeling framework captures $T_s$ variations across different elevations and geomorphic conditions more effectively than existing

datasets. However, the QTP and YGP are characterized by complex terrain and high altitudes, coupled with rapidly changing climatic conditions, which significantly complicate $T_s$ estimation. These findings align with previous studies showing that high elevations intensify the disconnect between air temperature and LST, thereby increasing the uncertainty in thermal modeling (Mo et al., 2025).

MODIS LST serves as a critical input to our modeling framework. However, as an optical remote sensing product, it is highly susceptible to cloud contamination, often resulting in data gaps. Despite the use of spatiotemporal interpolation and SG filtering, residual uncertainties persist in the reconstructed LST data. Future improvements in $T_s$ reconstruction can be pursued along two main directions. First, more physically grounded LST reconstruction methods can be adopted, such as incorporating surface energy balance models and diurnal temperature cycle models (Hong et al., 2022; Firozjaei et al., 2024; Wang et al., 2024). These methods apply energy conservation principles to estimate $T_s$ during periods of missing or unreliable observations, thereby providing more realistic estimates of land surface thermal conditions during periods of cloud cover. Second, integrating higher temporal resolution remote sensing observations may help overcome the limitations of MODIS. For instance, passive microwave satellite data provide all-weather observations and are less sensitive to cloud interference (Duan et al., 2017; Wu et al., 2022). In addition, next-generation geostationary satellites such as Himawari-8 offer observations at 10-minute intervals, substantially enhancing the temporal continuity and quality of surface temperature estimates (Yamamoto et al., 2022; You et al., 2024). These enhancements are expected to significantly improve the accuracy and temporal continuity of $T_s$ monitoring.

Our results (Figures 7 and 8) show that model accuracy varies across soil depths and is further influenced by season and land-use conditions. Accuracy is relatively lower at the surface (0 cm), improves at intermediate depths (5~10 cm), and declines again at deeper layers (20~40 cm). This depth-dependent pattern can be explained by the physical characteristics of the soil profile. Surface $T_s$ responds strongly to short-term meteorological fluctuations such as radiation, precipitation, and ET, resulting in

greater spatiotemporal variability and consequently larger prediction errors. In contrast, intermediate soil layers buffer high-frequency temperature fluctuations through thermal diffusion and higher heat capacity. As a result, $T_s$ becomes more stable with lower natural variability at these depths, leading to lower RMSE and higher R² values.

At deeper layers, prediction accuracy decreases because surface-level errors propagate downward through the hierarchical modeling framework, and uncertainties in soil texture inputs gradually accumulate with depth; during periods such as summer and winter, these combined uncertainties may be further amplified. Short-term changes in soil moisture alter fundamental soil thermal properties, including heat capacity,

thermal conductivity, and thermal diffusivity, which in turn control heat transfer processes and sub-daily $T_s$ dynamics. (Abu-Hamdeh, 2003; Subin et al., 2013). Consequently, the absence of soil moisture information may introduce additional uncertainty when modeling daily and sub-daily $T_s$ dynamics, especially at deeper layers. Incorporating high-resolution soil moisture datasets in future work would improve the

representation of soil hydrothermal interactions and further enhance $T_s$ estimation accuracy.

Seasonal variations and differences in land cover also contribute to the spatiotemporal differences in model performance. As shown in Figures 7 and 8, the model performs better in spring and autumn, whereas its accuracy declines in summer

and winter. In summer, vigorous vegetation growth and canopy closure alter surface–atmosphere energy exchange processes and weaken the relationship between canopy temperature and subsurface $T_s$, thereby reducing the effectiveness of LST as a proxy for near-surface $T_s$ (Kropp et al., 2020; Cui et al., 2022). Moreover, because satellite sensors measure radiometric temperature, LST in densely vegetated regions often represents

canopy-top temperature rather than the surface $T_s$, introducing an additional source of uncertainty. In winter, snow cover further increases complexity: the high albedo of snow reduces net radiation (Loranty et al., 2014; Li et al., 2018), and its insulating effect weakens the soil's response to cold-air fluctuations (Zhang, 2005; Myers-Smith et al., 2015). Meanwhile, Meanwhile, freezing of soil water alters soil thermal conductivity

and heat capacity, and frequent freeze–thaw cycles introduce nonlinear dynamics into $T_s$, increasing modeling uncertainty (Li et al., 2023a; Imanian et al., 2024). Although our multi-source adaptive modeling framework demonstrates robust performance across varying depths and environmental conditions, it does not explicitly represent the physical mechanisms governing vertical heat transfer. Future research could incorporate deep learning models capable of learning complex spatiotemporal dependencies to enhance the physical interpretability of $T_s$ variations across time, space, and depth.

## 5. Conclusion

This study addresses the lack of high spatiotemporal resolution multi-layer $T_s$ data by proposing a spatially adaptive ML framework, successfully constructing a retrieval model for multi-layer $T_s$. By integrating in-situ observations, reanalysis data, satellite remote sensing data, as well as topographic and soil texture data, the model demonstrates high accuracy across different depths, seasons, and land use types. The results indicate relatively higher performance in spring and autumn than in summer and winter, and greater accuracy in bare land, cropland, and grassland compared with forested areas. In comparison with ERA5-Land and GLDAS 2.1 $T_s$ products, the multi-layer $T_s$ data generated in this study exhibit significant improvements in both accuracy and spatial detail. Based on this framework, we have first developed the long-term (2010-2020) high spatiotemporal resolution (daily, 1 km resolution) multi-layer (0, 5, 10, 15, 20, 40 cm) $T_s$ dataset for China. Future research could further explore methods that simultaneously integrate temporal, spatial, and depth information, and utilize multi-source sensor data to enhance the spatiotemporal monitoring capabilities of $T_s$ at different depths. Overall, this study demonstrates the potential of multi-source data in $T_s$ estimation and provides a reliable tool and data foundation for ecological modeling, agricultural production and related studies.

## 6. Data availability

The daily multi-layer $T_s$ products (0, 5, 10, 15, 20, and 40 cm) at 1 km resolution from 2010 to 2020 are freely available in HDF5 format to the public at https://doi.org/10.11888/Terre.tpdc.302333 (Wang et al., 2025b). In addition, monthly multi-layer $T_s$ data are also provided to meet the needs of various users.

**7. Code availability**

The R scripts used to implement the rotated-quadtree spatial adaptive partitioning are publicly available at: https://github.com/wangxt1314/Rotated-quadtree

**Author contributions.** XW, JM and HS developed the methodology and designed the experiments. LH and XW collected and processed the data. XW wrote the first draft of the paper under the supervision of other authors. All authors participated in the review and editing of the paper.

**Competing interests.** Author Hao Shi is a member of the editorial board of Earth System Science Data. The contact author has declared that no other competing interests are present.

**Acknowledgments.** We gratefully acknowledge the National Meteorological Center of the China Meteorological Administration for providing the observed $T_s$ data. We thank the NASA Earth Observing System Data and Information System for providing MODIS and DEM data, and the European Centre for Medium-Range Weather Forecasts for the ERA5-Land reanalysis dataset. We also acknowledge the Soil SubCenter of the National Earth System Science Data Center, National Science & Technology Infrastructure of China (http://soil.geodata.cn), for providing soil texture data. The flux tower data used for independent validation were provided by the ChinaFLUX network through the National Ecosystem Science Data Center, National Science & Technology Infrastructure of China (http://www.nesdc.org.cn).

**Financial support.** This work was supported by the National Key Research and Development Program of China (Grant No. 2023YFF1303700) and the National
Natural Science Foundation of China (No. 42375195).

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
