# Peer review of "Spatially adaptive estimation of multi-layer soil temperature at a daily time-step"

_Earth System Science Data, 2025_

## Author Comment (AC1)

**Revisions of Manuscript: ESSD-2025-192**

**Title:** Spatially adaptive estimation of multi-layer soil temperature at a daily time-step across China during 2010-2020

**Author(s):** Xuetong Wang, Liang He, Peng Li, Jiageng Ma, Yu Shi, Qi Tian, Gang Zhao, Jianqiang He, Hao Feng, Hao Shi, Qiang Yu

Dear Reviewer,

We sincerely thank you for your thoughtful comments and constructive suggestions on our manuscript. We have carefully revised the manuscript in response to your feedback, with all changes clearly marked using track changes. In the revised manuscript and accompanying supplementary materials, modifications are highlighted in blue for ease of reference.

Below, we provide a detailed, point-by-point response to each of your comments. For clarity, your original remarks are shown in *italics*, followed by our corresponding replies. We have made every effort to address all concerns comprehensively and to improve the scientific rigor, clarity, and overall quality of the manuscript.

We sincerely appreciate the time and effort you invested in reviewing our work.

**Reviewer Comment 1:**

*In the introduction (lines 92–104), the authors clearly outline two key challenges in current research: first, the significant heterogeneity of $T_s$ leads to unclear relationships between variables; second, modeling is hindered by data scarcity and uneven distribution. However, in lines 106–113, when introducing the objectives and scope of this study, the authors do not explain how the study addresses these two challenges. It is also unclear what specific methods are used to overcome them, and why these methods are effective. It is recommended that the authors restructure this section by focusing on the core problems, rather than simply listing the research contents. This would improve the clarity and logical flow of the introduction.*

**Response to Reviewer Comment 1:**

We greatly appreciate your insightful comments and constructive feedback. We agree that the introduction should more explicitly link the identified challenges with the study's objectives and methodology. In response, we have revised and reorganized the relevant section to clarify how our approach directly addresses the two key challenges currently facing $T_s$ prediction.

**Revised Text (L105-L121):**

To address the above challenges, this study proposes a spatially adaptive methodology based on quadtrees. This approach dynamically partitions the study area into grids of varying sizes, with smaller grids in densely observed regions and larger grids in sparsely sampled areas, thereby enabling localized modeling that better captures spatial heterogeneity across complex environmental gradients. In addition, multi-source environmental predictors are integrated, and XGBoost models are applied within each grid cell to capture the nonlinear relationships between $T_s$ and its driving factors. Importantly, we employ a spatial block cross-validation strategy to evaluate the model's generalization ability in unseen regions. Based on this framework, the objectives of this study are to: (1) construct a spatially adaptive modeling system; (2) generate a multi-layer $T_s$ dataset at a daily time-step and one kilometer resolution in China from 2010-2020; and (3) evaluate the dataset through independent validation with flux tower observations and benchmarking against widely used $T_s$ products. The proposed methodology could directly address the scaling challenges induced by spatial heterogeneity and uneven data distribution. The generated products would provide a robust foundation for high-resolution environmental modeling, precision agriculture and climate impact assessments.

**Reviewer Comment 2:**

*In Section 2.1, the authors describe the use of CMA $T_s$ observational data. However, it is unclear how these data were processed. Were the observations directly provided as daily averages, or were they aggregated from hourly data? Was any quality control applied? How were missing data handled, both in the vertical profile and in the time series? Were any filtering or screening steps performed, and if so, what were the specific criteria?*

**Response to Reviewer Comment 2:**

We appreciate the reviewer's thoughtful comment. The multi-layer $T_s$ data were obtained from the national CMA weather station network, where measurements were automatically recorded every 10 minutes and used to compute daily means at each depth. Data preprocessing steps are described in Section 2.1.

**Revised Text (L125-L133):**

In this study, in-situ $T_s$ *observations* was measured at six depths: at the surface (0 m), and at subsurface levels of 0.05, 0.10, 0.15, 0.20, and 0.40 meters. Data were collected through the national weather station network operated by the China Meteorological Administration (CMA), in accordance with standardized measurement protocols. At each site, $T_s$ was recorded every 10 minutes and automatically uploaded to a central server. Daily mean values at each depth were calculated from these high-frequency records. We then assessed data completeness for the period 2010–2020 and excluded stations with more than 20% missing daily records at any depth. After quality control, 2,093 stations were retained for model development.

**Reviewer Comment 3:**

*In lines 186–190, as well as in Section 4.3, the authors provide a brief discussion of the study's limitations. However, it is concerning that the missing land surface temperature (LST) data caused by cloud cover were filled using a simple linear interpolation method. This approach may be questionable, as the interpolated values represent a theoretical cloud-free state, while cloud presence can significantly influence radiative transfer and thus impact LST. There are existing interpolation methods that take into account energy transfer and energy balance. It is recommended that the authors investigate these alternatives and consider adopting a more reliable method.*

**Response to Reviewer Comment 3:**

We appreciate your comments on the interpolation method used to address LST gaps resulting from cloud contamination. Indeed, cloud cover presents a major challenge in remote sensing–based LST reconstruction, as it significantly alters surface radiative fluxes and interferes with the physical basis of thermal observations. As the reviewer correctly noted, linear interpolation does not explicitly account for the thermal effects of clouds and may produce overly idealized estimates under cloud-free assumptions.

In this study, we employed a spatiotemporal linear interpolation method primarily due to its computational efficiency, simplicity, and suitability for large-scale reconstruction of missing data. To further reduce short-term fluctuations and noise introduced during interpolation, we applied a Savitzky–Golay filter during the preprocessing stage to smooth the time series (Kong et al., 2019; Chen et al., 2021). Notably, this method can be readily implemented on the Google Earth Engine (GEE) platform, enabling efficient global processing of MODIS LST products and the rapid generation of daily gap-free land surface temperature composites. This facilitates scalable model training and $T_s$ estimation.

Nevertheless, we fully acknowledge the limitations of this method in cases of prolonged cloud cover. We concur that incorporating physically based interpolation methods could enhance the reliability of the reconstructed data. In future work, we plan to explore energy balance–based reconstruction techniques, such as incorporating surface energy balance system models and diurnal temperature cycle models (Hong et al., 2022; Firozjaei et al., 2024; Wang et al., 2024).

Moving forward, we aim to explore hybrid approaches that combine physically based models with machine learning algorithms to better capture the effects of cloud cover, land surface heterogeneity, and seasonal variability on $T_s$ reconstruction. Additionally, we intend to incorporate passive microwave–based land surface temperature products, which are less affected by cloud contamination, as supplementary information for gap-filling. We believe these advancements will help reduce uncertainties in LST reconstruction and further enhance the accuracy and robustness of the resulting $T_s$ dataset.


We sincerely thank the reviewer for the insightful comments regarding the design rationale of the quadtree-based partitioning strategy. To justify our choice of threshold = 30 as the final splitting criterion, we conducted a systematic evaluation of the partitioning performance under different thresholds using three key metrics. The supporting analysis and figures are included in the Appendix.

**Here are the revisions, supplemented in the Appendix (L14-L44):**

We conducted a systematic evaluation of the partitioning performance under different thresholds using three key metrics: the coefficient of variation (CV) of point count, the CV of point density, and the total number of grid cells. The CV of point count was used to evaluate the balance of sample distribution across spatial units under different thresholds. Point density was defined as the number of observation stations within a grid cell divided by its area. A lower CV of point density indicates that the partitioning effectively adjusted grid size according to local station density—i.e., producing smaller grids in dense regions and larger grids in sparse areas—thus reflecting a more adaptive spatial division. Conversely, a higher CV suggests that the partitioning failed to capture the spatial heterogeneity of station density. Therefore, the CV of point density serves as a key indicator of the spatial adaptivity of the quadtree partitioning.

The total number of grids corresponds to the number of local models to be trained, and thus indirectly reflects the computational and time cost associated with model training. As shown in Figure S4 (a–c), we systematically evaluated quadtree performance under a series of point-count thresholds (10, 30, 50, 70, 90): Figure S4a shows that the CV of point count drops rapidly with increasing threshold, indicating improved balance in

sample allocation across grids. However, this trend levels off beyond threshold = 30, suggesting diminishing returns. Thus, threshold 30 marks an optimal trade-off. Figure S4b shows a notable inflection point in the CV of point density near threshold = 30. Although not the global minimum, this point represents an optimal trade-off where grid subdivision sufficiently reflects sample density variation without causing over- or under-segmentation—thereby capturing spatial adaptivity effectively. Figure S4c shows that the number of grid cells decreases rapidly as the threshold increases, leading to substantial computational savings. However, the rate of reduction slows considerably beyond threshold = 30, indicating limited additional benefit from further increases.

In summary, threshold = 30 achieves a favorable balance among sample distribution equity, spatial adaptivity, and computational efficiency, and was therefore selected as the final splitting threshold in this study. The detailed results of this threshold evaluation, including figures and metric comparisons, have been added to the revised manuscript as supplementary material (Appendix, Lines 9–41) to support transparency.

[Figure]

**Figure S4.** Performance evaluation of quadtree partitioning under different point-count thresholds. (a) Coefficient of variation (CV) of point count across spatial units. (b) CV of point density (point count per unit area). (c) Total number of generated grid cells. Dashed vertical line indicates the selected threshold of 30.

**3. Comparison with the K-D Tree Approach**

We appreciate the reviewer's thoughtful suggestion regarding the use of K-D trees for achieving a balanced spatial distribution of stations. We agree that K-D trees offer precise control over sample counts in each partition and can be advantageous when strict sample balance is the primary objective. However, the core objective of our study is not to enforce equal sample sizes in each spatial unit, but rather to enhance the adaptability and predictive performance of local modeling under spatially heterogeneous station distributions. To this end, we adopted a bottom-up quadtree-based strategy, which recursively subdivides space based on a point-count threshold. This enables the generation of finer grids in data-rich areas and larger cells in sparse regions, allowing the model structure to adapt to local data density and environmental variability. Compared to top-down methods like K-D trees, the quadtree is better suited for capturing spatial adaptivity than enforcing uniform sample counts. That said, we

acknowledge the merits of K-D trees and agree that they represent a promising alternative for future work, particularly in applications where sample balance is more critical than spatial adaptivity.

**Reviewer Comment 6:**

*In Section 2.3.3 (lines 285–295), the authors introduce XGBoost as the core machine learning algorithm used in the study. They present its advantages and compare it with other methods such as SVM, RF, and neural networks. However, the stated advantages are not sufficient to demonstrate that XGBoost is superior to the other listed methods. Machine learning models differ in structure, number of parameters, optimization strategy, and suitability for different tasks. Therefore, the current explanation is not enough to justify the model choice. Considering that the algorithm is not the main focus of this paper, it is suggested to either include a brief comparative experiment to support the claimed superiority or rephrase the section to emphasize the strengths of XGBoost without direct comparison to other models.*

**Response to Reviewer Comment 6:**

We appreciate the reviewer's constructive comments regarding the justification of our model choice. As suggested, to clarify our reasoning, we have elaborated on the key considerations below.

**Revised Text (L303-L317):**

We adopted the XGBoost (Extreme Gradient Boosting) algorithm as the core regression model for $T_s$ estimation due to its strong predictive performance, computational efficiency, and scalability across large environmental datasets. XGBoost builds an ensemble of regression trees in a stage-wise boosting process, where each tree is trained to minimize the residuals from the previous iteration, leading to a robust and optimized model (Chen and Guestrin, 2016). A key strength of XGBoost is its ability to handle heterogeneous and high-dimensional predictor sets, which are common in geoscience applications involving complex terrain, land cover variability, and climatic gradients. Recent studies have demonstrated its effectiveness in similar domains, including land surface temperature reconstruction (Li et al., 2024), multi-layer soil moisture estimation (Karthikeyan and Mishra, 2021), drought event attribution (Wang et al., 2025), and crop yield prediction (Li et al., 2023b). Given these proven strengths and the spatially nonstationary characteristics of $T_s$ in our study area, XGBoost was selected to train localized prediction models within spatial subregions.


**Reviewer Comment 7:**

*In lines 296–297, the validation set is twice the size of the test set. Is this split reasonable, and can it effectively evaluate the generalization performance of the model? Why not adopt more common ratios such as 8:1:1 or 6:2:2? In addition, the manuscript later mentions that five-fold cross-validation was used for evaluation. In this context, what are the roles of the two validation sets? Are they used for model selection, parameter tuning, or testing? It is recommended that the authors provide a clearer explanation. It is also suggested to report the specific sample sizes for each dataset.*

**Response to Reviewer Comment 7:**

We thank the reviewer for this valuable comment. In the revised manuscript, we have refined the data partitioning strategy and provided a clearer explanation of the roles of each dataset.

Specifically, to rigorously evaluate the spatial generalization performance of the model and avoid potential data leakage, we employed spatial block cross-validation combined with GridSearchCV during localized modeling. In this method, observation sites were first grouped into spatial blocks based on their geographic locations, and cross-validation was then conducted across blocks rather than through random splitting at the individual site level. This approach ensured that geographically adjacent sites were not simultaneously included in both the training and testing subsets, thereby enabling a stricter and more realistic assessment of the model's generalization ability to new regions. Based on this revised scheme, we retrained and re-evaluated the XGBoost models. The updated results and methodological details are now presented in the revised manuscript (L318–336).

As this study involves multiple soil depths and spatial subregions, the exact sample sizes vary across cases and are therefore not reported individually in the main text.

However, we have clearly specified the data partitioning ratios and their purposes to ensure methodological transparency and reproducibility. We believe that this revised scheme not only aligns with common practice but also provides a stricter and more realistic evaluation of the model's generalization performance.

**Revised Text (L318-336):**
To rigorously account for the strong spatial autocorrelation of $T_s$ and avoid potential data leakage between training and testing subsets, we employed a spatial block cross-validation scheme rather than random splitting. Specifically, within each rotated quadtree grid, observation sites were grouped into spatial blocks based on their geographic coordinates: station latitude and longitude were each divided by 1° and floored to integer values, and stations sharing the same index were assigned to the same block. This ensured that samples within the same spatial block were not simultaneously assigned to both the training and testing subsets, thereby avoiding data leakage due to spatial autocorrelation and enabling a more reliable evaluation of the model's generalization capability.

Within each spatial grid, the data were partitioned into training (90%) and testing (10%) subsets at the block level. The training subset was further subjected to 10-fold spatial block cross-validation using GridSearchCV to optimize three key hyperparameters: the number of trees (n_estimators), maximum tree depth (max_depth), and learning rate (learning_rate). Detailed parameter settings are provided in Appendix Table S1. The hyperparameter set that yielded the lowest average validation error across the ten folds was selected as optimal. The final model was retrained on the full training set with the optimized parameters and evaluated on the held-out testing set to assess generalization.

**Reviewer Comment 8:**
*The authors produced data at a 1-kilometer resolution for China. How did the authors account for the spatial scale difference between point observations of soil temperature and the 1-kilometer resolution results? How was it ensured that the dataset constructed through point observation training could represent results at the 1-kilometer spatial scale? Additionally, regarding the dataset production, I am very interested in the subsequent maintenance and updates of the dataset over time. Can the authors' method be extended to produce datasets for subsequent years?*

**Response to Reviewer Comment 8:**
We thank the reviewer for this important and thoughtful comment. It involves two critical aspects:
(1) the scale consistency between point-based observations and gridded predictions at a 1 km resolution;
(2) the potential for dataset maintenance and future updates. We address both issues below.

**1. Addressing the Scale Difference Between Point Observations and 1 km Predictions**

To reconcile the spatial scale mismatch between point-level $T_s$ observations and the 1 km gridded outputs, we implemented a multi-pronged modeling strategy designed to ensure scale compatibility and representativeness:

(1) Predictor Resolution consistency:

All input variables used for model training (e.g., MODIS, ERA5-Land, and soil texture data) were uniformly resampled to a spatial resolution of 1 kilometer, thereby ensuring that the spatial scale of the predictors is consistent with that of the target output.

(2) Rotated Quadtree-Based Local Modeling:

As detailed in the revised Section 2.3.2, we employed a spatially adaptive modeling strategy based on rotated quadtree partitioning. This approach automatically divides the study area into spatial units of varying sizes according to the density of observation stations—finer grids in densely sampled areas and coarser grids in sparsely observed regions. Within each unit, a localized XGBoost model was trained using in-situ observations and 1 km-resolution environmental predictors. To mitigate edge effects and directional bias introduced by fixed partition boundaries, we constructed quadtree structures under six different rotation angles (0° to 75°). For each soil depth layer, the predictions from these rotated models were averaged, thereby reducing boundary artifacts and enhancing the spatial continuity and robustness of the final results.

(3) Robust Evaluation Framework:

A two-tier validation framework was established to comprehensively assess model performance. First, we applied spatial block cross-validation within each rotated quadtree grid. In this scheme, observation sites were partitioned into training (90%) and testing (10%) subsets at the block level, ensuring that geographically adjacent sites were not simultaneously included in both subsets. The training subset was further subjected to 10-fold cross-validation for parameter tuning, while the testing subset was used to rigorously evaluate spatial generalization. This approach effectively reduced the risk of data leakage caused by spatial autocorrelation and enhanced the robustness of the evaluation. Second, independent external validation was performed using daily $T_s$ observations from 18 flux tower sites of the ChinaFLUX network. The results (Section 3.1, Figure 5) show that the dataset maintains high accuracy at these independent sites, further confirming the reliability and robustness of the evaluation framework.

(4) Established Precedents:

The use of point-based observations to train models for gridded prediction has been widely applied in related environmental studies, such as land surface temperature and soil moisture estimation (Karthikeyan and Mishra, 2021; Song et al., 2022; Yu et al., 2024). Our method builds on these established practices by incorporating spatial adaptivity and ensemble averaging, further enhancing consistency and robustness.

**2. Potential for Dataset Extension and Future Updates**

We greatly appreciate the reviewer's interest in the extensibility and long-term value of the dataset. As elaborated in the revised discussion section, the proposed spatially adaptive modeling framework is designed to be modular and scalable, making it readily

applicable to future years. Given access to updated in-situ station observations and corresponding environmental predictors (e.g., MODIS and ERA5-Land), the same modeling pipeline can be re-applied to retrain the models and generate new products. This allows for filling historical data gaps and extending $T_s$ estimates into future periods. In addition, we are currently generating $T_s$ estimates for the period 2001–2010, which will soon be released through the National Tibetan Plateau Data Center (https://data.tpdc.ac.cn). Beyond this, the dataset will be continuously maintained and updated, with all future versions openly released on the same platform to ensure free and unrestricted access for the global scientific community. We believe these ongoing efforts will provide long-term benefits for environmental monitoring, climate research, and ecosystem modeling.

We sincerely thank the reviewer for raising the important issue of uneven spatial distribution of $T_s$ observation sites and its implications for national-scale dataset development. As noted, $T_s$ stations in China are concentrated in eastern lowland regions, with sparse coverage across the western and high-altitude areas. This spatial imbalance poses a major challenge to constructing a robust and spatially representative $T_s$ dataset.

To address this, we adopted a spatially adaptive modeling framework based on rotated quadtree partitioning approach. This method improves the dataset construction in two primary ways. First, it dynamically subdivides the study area into spatial units based on observation density: finer grids are assigned to densely sampled regions to improve local precision and avoid overfitting, while coarser grids are used in sparsely sampled

areas to maintain model stability and statistical representativeness. Within each grid cell, a localized XGBoost model is trained to capture nonlinear relationships between $T_s$ and relevant environmental drivers, including topography, landforms, climate, and vegetation. This strategy mitigates structural biases associated with training a single global model on unevenly distributed data. Second, to reduce boundary artifacts caused by fixed grid divisions, we generated quadtree structures under multiple rotation angles and averaged their predictions. This ensemble strategy enhanced the spatial coherence and robustness of the final $T_s$ dataset (see revised Section 2.3 and Section 4.1 for detailed explanations).

[revised manuscript text omitted]

**2. Influence of Topography, Climate, and Vegetation on Model Performance and Uncertainty**

We also thank the reviewer for pointing out the potential influence of environmental factors on model uncertainty. As shown in Sections 3.2 and 3.3 of the revised manuscript, although the overall accuracy of the dataset is satisfactory, the estimation performance exhibits clear spatial and seasonal heterogeneity. To address this, we expanded the discussion in Section 4.3 to systematically examine how factors such as topography, climate conditions, land cover types, and remote sensing variables may affect the stability and accuracy of $T_s$ estimates across different regions and seasons. We also proposed future directions for improving model adaptability under complex environmental conditions. These revisions aim to clarify how our methodology accounts for spatial sampling bias and environmental complexity, and we hope they address the reviewer's concerns comprehensively.

**Revised Text (L602-L662):**

[revised manuscript text omitted]

**Reviewer Comment 10:**

*There are also some minor issues that should be addressed. For example, in Figures 6 and 7, it is recommended to include a color bar legend. As it stands, it is difficult to interpret the exact values represented by the orange points. In Equation (2), the variables x and y lack subscripts i. In Equation (4), the variable i used for summation is not defined. In the references, line 667 and 763 include "others" among the authors— what does this mean? It is suggested to carefully check the manuscript for such details, including grammar, figures, and reference formatting.*

**Response to Reviewer Comment 10:**

We thank the reviewer for the careful reading and helpful suggestions. In response:

**1. Figure revisions**

We have redrawn the portion of Figure 1 related to dataset division in the revised manuscript to present it more clearly to the readers. Additionally, we have added color bar legends to both Figure 6 and Figure 7. This addition clarifies the exact values represented by the orange points and enhances the interpretability of the figures.

**Revised Text (L342-L344):**

[Figure]

**Figure 3.** Workflow of the proposed method to obtain multi-layer $T_s$ over the China.

**Revised Text (L402-L408):**

[Figure]

**Figure 6.** Goodness of R² across China estimated during the model testing phase. Performance metrics are calculated between predicted_$T_s$ and in-situ $T_s$ data sets.

[Figure]

**Figure 7.** Goodness of RMSE across China estimated during the model testing phase. Performance metrics are calculated between predicted $T_s$ and in-situ $T_s$ data sets.

**2. Equation corrections**

**Revised Text (L352-L359):**
Equation (2) has been corrected to include subscripts i for both x and y, to clearly indicate that the RMSE is calculated over paired observations.

$$RMSE = \sqrt{\frac{\sum_{i=1}^{N}[(x_i - \overline{X}) - (y_i - \overline{Y})]^2}{N}} \tag{1.2}$$

Equation (4) has been reformulated to explicitly define the summation index i and to reflect the mean bias across all samples.

$$Bias = \frac{1}{N}\sum_{i=1}^{N}(x_i - y_i) \tag{1.3}$$

**3. Reference formatting**
In accordance with the reviewer's suggestion, we have carefully reviewed and revised the entire reference list to ensure formatting accuracy and consistency, fully complying with the journal's citation requirements.

**4. Additional Edits**
We carefully reviewed the manuscript to address minor issues in grammar, figure annotations, and reference formatting. We are grateful for the reviewer's attention to these important details, which helped us further improve the overall clarity and quality of the manuscript.

---

## Author Comment (AC2)

**Revisions of Manuscript:  ESSD-2025-192**

**Title:** Spatially adaptive estimation of multi-layer soil temperature at a daily time-step across China during 2010-2020

**Author(s):** Xuetong Wang, Liang He, Peng Li, Jiageng Ma, Yu Shi, Qi Tian, Gang Zhao, Jianqiang He, Hao Feng, Hao Shi, Qiang Yu

Dear Reviewer,

We sincerely thank you for your thoughtful comments and constructive suggestions on our manuscript. We have carefully revised the manuscript in response to your feedback, with all changes clearly marked using track changes. In the revised manuscript and accompanying supplementary materials, modifications are highlighted in blue for ease of reference.

Below, we provide a detailed, point-by-point response to each of your comments. For clarity, your original remarks are shown in *italics*, followed by our corresponding replies. We have made every effort to address all concerns comprehensively and to improve the scientific rigor, clarity, and overall quality of the manuscript.

We sincerely appreciate the time and effort you invested in reviewing our work, and we believe the revisions have significantly improved the manuscript.

**Reviewer Comment 1:**

*The authors state that the sites were randomly split into training (70%), validation (20%), and test (10%) sets. For geospatial data like soil temperature, which exhibits strong spatial autocorrelation, this random splitting is a critical methodological flaw. It almost certainly leads to "data leakage", where test sites are geographically close to training sites. Consequently, the model can achieve high performance on the test set even though it did not learn the true underlying relationships between predictors and Ts. This means the model's ability to generalize to new, un-sampled areas is not being properly evaluated. The reported performance metrics (e.g., R² > 0.93 in Fig. 5) are therefore very likely to be significantly inflated and overly optimistic.*

*The authors should implement a more rigorous validation scheme that accounts for spatial autocorrelation. A spatial block cross-validation approach is strongly recommended.*

**Response to Reviewer Comment 1:**

We sincerely thank the reviewer for this insightful comment. We fully agree that random splitting of sites into training, validation, and test sets may lead to spatial data leakage due to the strong spatial autocorrelation of soil temperature. This could indeed result in overly optimistic performance metrics and an inaccurate assessment of the model's spatial generalization ability.

In response, we have revised our methodology by adopting a spatial block cross-validation scheme to partition the data. Specifically, observation sites were grouped into spatial blocks, and the cross-validation was conducted across these blocks rather than through random splits. This approach ensures that geographically adjacent sites are not simultaneously included in both training and testing subsets, thereby providing a more rigorous and realistic evaluation of model generalization to un-sampled regions.

We have retrained and re-evaluated the XGBoost models using this revised validation strategy. The updated results, along with a detailed description of the method, are now presented in the manuscript.

**Revised Text (L318–336):**

To rigorously account for the strong spatial autocorrelation of $T_s$ and avoid potential data leakage between training and testing subsets, we employed a spatial block cross-validation scheme rather than random splitting. Specifically, within each rotated quadtree grid, observation sites were grouped into spatial blocks based on their geographic coordinates: station latitude and longitude were each divided by 1° and floored to integer values, and stations sharing the same index were assigned to the same block. This ensured that samples within the same spatial block were not simultaneously assigned to both the training and testing subsets, thereby avoiding data leakage due to spatial autocorrelation and enabling a more reliable evaluation of the model's generalization capability.

Within each spatial grid, the data were partitioned into training (90%) and testing (10%) subsets at the block level. The training subset was further subjected to 10-fold spatial block cross-validation using GridSearchCV to optimize three key hyperparameters: the number of trees (n_estimators), maximum tree depth (max_depth), and learning rate (learning_rate). Detailed parameter settings are provided in Appendix Table S1. The hyperparameter set that yielded the lowest average validation error across the ten folds was selected as optimal. The final model, retrained on the full training set with these parameters, was then evaluated on the held-out testing blocks to assess its generalization ability and examine potential overfitting within each grid.

**Reviewer Comment 2:**

*The manuscript claims that the generated dataset accurately captures the spatial distribution of Ts. However, the evidence provided is the high R² (and low RMSE) of the daily time series at individual stations. These temporal variations are heavily dominated by the seasonal cycle, which is easy for any model to capture using predictors like air temperature. A high temporal R² does not prove that the model correctly reproduces the spatial gradients across China. I suggest the authors conduct a spatial-only validation, using mean Ts (for the whole year and for specific seasons) across the sites.*

**Response to Reviewer Comment 2:**

We thank the reviewer for this constructive suggestion. We agree that high temporal $R^2$ at individual stations mainly reflects the ability to capture seasonal variations and may not sufficiently demonstrate the model's capacity to reproduce spatial gradients. Following the reviewer's advice, we conducted a spatial-only validation using annual mean $T_s$ across all stations. The results are presented in Figure 1, which compares the estimated and observed annual mean $T_s$ at depths from 0–40 cm. Each point represents the annual mean $T_s$ at a single site. The results indicate high correlations ($R^2 = 0.995$–$0.997$) and low errors (RMSE = 0.26–0.37 K; MAE = 0.19–0.28 K), demonstrating that the generated dataset reliably captures the spatial distribution of $T_s$ across sites. These additional analyses provide strong evidence that our dataset reproduces both temporal dynamics and spatial gradients of $T_s$ across China.

[Figure]

**Figure 1.** Validation of spatial patterns of annual mean $T_s$ at different soil depths across China.

**Reviewer Comment 3:**
*The results indicate that model performance is worse at the surface (0 cm) and improves at intermediate depths (e.g., 5-20 cm), as shown in Figures 4-7. This is a counter-intuitive result given the layer-cascading methodology, where the prediction for a deeper layer depends on the prediction from the layer above. This structure implies that errors from the surface prediction should propagate downwards, theoretically leading to a degradation of performance with depth. This apparent paradox should be discussed.*

**Response to Reviewer Comment 3:**
As the reviewer correctly noted, our revised modeling results reveal clear depth-dependent variations in prediction accuracy. Overall, acceptable performance was achieved across all depths. Errors were relatively larger at the 0 cm surface layer, whereas predictions at 5 cm and 10 cm depths showed improved accuracy compared to the surface. With further increases in depth (20–40 cm), errors tended to accumulate, and this pattern was particularly evident in summer and winter.

This phenomenon can be explained by the physical characteristics of soil temperature dynamics. The surface layer is strongly influenced by high-frequency environmental disturbances such as radiation, precipitation, and evapotranspiration, which elevate the noise level and complicate accurate prediction. In contrast, intermediate layers benefit from the buffering effects of thermal diffusion and soil heat capacity, which dampen short-term fluctuations and make temperature variations more stable and thus more predictable. At greater depths, however, cascading errors are gradually propagated and

amplified, resulting in reduced accuracy. We have revised the manuscript to include a detailed discussion on the rationale behind this result.

**Revised Text (L632-662):**

Our results (Figures 8 and 9) show that model accuracy varies across different soil depths, with additional influences from season and land use. Accuracy is relatively lower at the surface (0 cm), improves at intermediate depths (5–10 cm), and then declines again at greater depths (20–40 cm). This depth-dependent pattern can be explained by the physical characteristics of soil temperature. Surface soil temperature is highly sensitive to short-term meteorological fluctuations such as radiation, precipitation, and evapotranspiration, leading to greater spatiotemporal variability and larger prediction errors. In contrast, intermediate soil layers benefit from the buffering effects of thermal diffusion and soil heat capacity, which dampen high-frequency fluctuations and stabilize the relationship between predictors and $T_s$, thereby improving performance at these depths. At greater depths, however, surface-level errors propagate downward through the cascading framework, resulting in reduced accuracy—particularly during summer and winter.

Seasonal changes and variations in land cover further contribute to differences in estimation accuracy. As shown in Figures 8 and 9, the model exhibits higher accuracy in spring and autumn, whereas its performance tends to decline during summer and winter. During summer, dense vegetation growth and canopy closure reduce the influence of surface–atmosphere energy exchanges on $T_s$, weakening the correlation between canopy temperature and subsurface $T_s$ (Kropp et al., 2020; Cui et al., 2022). In winter, snow cover introduces a suite of confounding effects: high surface albedo reduces net radiation (Loranty et al., 2014; Li et al., 2018), while snow acts as an insulator, limiting the soil's response to cold air incursions (Zhang, 2005; Myers-Smith et al., 2015). Additionally, low temperatures lead to soil water freezing, which alters the soil's thermal conductivity and heat storage capacity. These factors, together with frequent freeze–thaw cycles, introduce complex nonlinear dynamics in $T_s$ that increase modeling uncertainty (Li et al., 2023a; Imanian et al., 2024). While our multi-source adaptive modeling framework performs well across depths, it does not explicitly account for the physical mechanisms of vertical heat transfer. Future research could explore deep learning models that are capable of learning complex spatiotemporal features and improving the physical interpretability of $T_s$ variations across time, space, and depth.

**Reference**

Cui, X., Xu, G., He, X., and Luo, D.: Influences of seasonal soil moisture and temperature on vegetation phenology in the Qilian Mountains, Remote Sens., 14, 3645, https://doi.org/10.3390/rs14153645, 2022.

Imanian, H., Mohammadian, A., Farhangmehr, V., Payeur, P., Goodarzi, D., Hiedra Cobo, J., and Shirkhani, H.: A comparative analysis of deep learning models for

soil temperature prediction in cold climates, Theor. Appl. Climatol., 155, 2571–2587, https://doi.org/10.1007/s00704-023-04781-x, 2024.

Kropp, H., Loranty, M. M., Natali, S. M., Kholodov, A. L., Rocha, A. V., Myers-Smith, I., Abbot, B. W., Abermann, J., Blanc-Betes, E., Blok, D., Blume-Werry, G., Boike, J., Breen, A. L., Cahoon, S. M. P., Christiansen, C. T., Douglas, T. A., Epstein, H. E., Frost, G. V., Goeckede, M., Høye, T. T., Mamet, S. D., O'Donnell, J. A., Olefeldt, D., Phoenix, G. K., Salmon, V. G., Sannel, A. B. K., Smith, S. L., Sonnentag, O., Vaughn, L. S., Williams, M., Elberling, B., Gough, L., Hjort, J., Lafleur, P. M., Euskirchen, E. S., Heijmans, M. M., Humphreys, E. R., Iwata, H., Jones, B. M., Jorgenson, M. T., Grünberg, I., Kim, Y., Laundre, J., Mauritz, M., Michelsen, A., Schaepman-Strub, G., Tape, K. D., Ueyama, M., Lee, B.-Y., Langley, K., and Lund, M.: Shallow soils are warmer under trees and tall shrubs across arctic and boreal ecosystems, Environ. Res. Lett., 16, 015001, https://doi.org/10.1088/1748-9326/abc994, 2020.

Li, Q., Ma, M., Wu, X., and Yang, H.: Snow cover and vegetation-induced decrease in global albedo from 2002 to 2016, J. Geophys. Res. Atmospheres, 123, 124–138, https://doi.org/10.1002/2017JD027010, 2018.

Li, X., Zhu, Y., Li, Q., Zhao, H., Zhu, J., and Zhang, C.: Interpretable spatio-temporal modeling for soil temperature prediction, Front. For. Glob. Change, 6, 1295731, https://doi.org/10.3389/ffgc.2023.1295731, 2023.

Loranty, M. M., Berner, L. T., Goetz, S. J., Jin, Y., and Randerson, J. T.: Vegetation controls on northern high latitude snow-albedo feedback: Observations and CMIP 5 model simulations, Glob. Change Biol., 20, 594–606, https://doi.org/10.1111/gcb.12391, 2014.

Myers-Smith, I. H., Elmendorf, S. C., Beck, P. S. A., Wilmking, M., Hallinger, M., Blok, D., Tape, K. D., Rayback, S. A., Macias-Fauria, M., Forbes, B. C., Speed, J. D. M., Boulanger-Lapointe, N., Rixen, C., Lévesque, E., Schmidt, N. M., Baittinger, C., Trant, A. J., Hermanutz, L., Collier, L. S., Dawes, M. A., Lantz, T. C., Weijers, S., Jørgensen, R. H., Buchwal, A., Buras, A., Naito, A. T., Ravolainen, V., Schaepman-Strub, G., Wheeler, J. A., Wipf, S., Guay, K. C., Hik, D. S., and Vellend, M.: Climate sensitivity of shrub growth across the tundra biome, Nat. Clim. Change, 5, 887–891, https://doi.org/10.1038/NCLIMATE2697, 2015.

Zhang, T.: Influence of the seasonal snow cover on the ground thermal regime: An overview, Rev. Geophys., 43, https://doi.org/10.1029/2004RG000157, 2005.

**Reviewer Comment 4:**

*In the VIF analysis (Fig. S1), sand, silt, and clay percentages were included. As these three variables are compositional and should sum to a constant (100%), they are perfectly collinear by definition. This should result in an infinite (or extremely large) VIF values. However, the reported VIFs are relatively low (5.6 to 10). This discrepancy is concerning and suggests a methodological error.*

**Response to Reviewer Comment 4:**

We thank the reviewer for pointing out this important issue. The reviewer is correct that sand, silt, and clay percentages are compositional variables that sum to 100% and are therefore perfectly collinear by definition. When variables are perfectly collinear, VIF cannot be correctly computed, as the underlying regression matrix becomes singular. Including all three variables simultaneously in the VIF analysis was therefore inappropriate, and we acknowledge that this led to misleading values (5.6–10) instead of extremely high or infinite VIFs.

In the revised manuscript, we have addressed this issue by excluding silt from the VIF analysis, since the three variables contain redundant information. This adjustment removes perfect collinearity and allows the VIF analysis to be correctly applied. The updated VIF results are now reported in the Supplementary Material (Fig. S2), and the corresponding text has been revised accordingly.

We further retrained the XGBoost models using the revised set of predictor variables and a spatial block cross-validation data partitioning strategy, and regenerated new data products to ensure the consistency and robustness of the analysis results. We sincerely appreciate the reviewer's suggestion, which has enabled us to improve the methodological rigor and reliability of our study.

[Figure]

**Figure S2.** Variance Inflation Factor (VIF) of predictor variables

**Revised Text (L244-254):**

Multicollinearity among multiple source variables may affect the robustness of the models. Therefore, we rigorously evaluated the multicollinearity among the independent variables using the variance inflation factor (VIF) before modeling to remove highly correlated variables. The VIF is a diagnostic statistic used to quantify the degree of multicollinearity by measuring how much the variance of a regression coefficient is inflated due to correlations with other predictors (Akinwande et al., 2015). It is calculated as:

$$VIF_i = \frac{1}{1 - R_i^2} \tag{1}$$

where $R_i^2$ is the coefficient of determination obtained by regressing the $i$-th predictor against all other predictors. Variables with VIF exceeding 10 are generally considered severely multicollinear and should be removed.

**Reference**

Akinwande, M. O., Dikko, H. G., and Samson, A.: Variance inflation factor: As a condition for the inclusion of suppressor variable(s) in regression analysis, Open J. Stat., 5, 754–767, https://doi.org/10.4236/ojs.2015.57075, 2015.

**Reviewer Comment 5:**

*The model uses solar radiation as a predictor but omits downward longwave radiation (LWD). Considering that LWD is a critical driver of the surface energy balance*

*(particularly for nighttime and winter temperatures) and that LWD has been identified as a main driver of Ts trends in process-based models (Peng et al., 2016, https://doi.org/10.5194/tc-10-179-2016), I suggest the authors include LWD as a predictor, or provide a strong justification for its exclusion.*

**Response to Reviewer Comment 5:**
We thank the reviewer for this valuable suggestion. Following the reviewer's comment, we incorporated downward longwave radiation (LWD) from ERA5 as a candidate predictor and evaluated its multicollinearity with other variables. The analysis revealed that LWD is highly collinear with solar radiation (revised Fig. S1). Considering that our study focused on daily mean $T_s$, the additional contribution of LWD was limited at the daily scale, as its effect on the surface energy balance was already largely captured by solar radiation. For these reasons, we excluded LWD from the final modeling to avoid redundancy and potential instability in the regression framework. Importantly, the inclusion or exclusion of LWD did not materially change the results or conclusions of our study.

This clarification has been added to the revised manuscript, and the updated figure illustrating the collinearity analysis is provided in the Supplementary Material.

**Revised Text (L158-164):**
In addition, both net solar radiation and downward longwave radiation (LWD) were considered. Net solar radiation directly represents the shortwave energy absorbed by the land surface and serves as the primary driver of the daytime surface energy budget, whereas LWD plays a particularly important role under nighttime and winter conditions by regulating surface heat loss through the longwave radiation balance. Together, they jointly control the surface energy balance and directly drive the spatiotemporal dynamics of $T_s$ (Peng et al., 2016).

**Revised Text (L255-268):**
Based on the VIF analysis, we applied the following adjustments to the predictor set. Accordingly, some variables were excluded due to severe multicollinearity or redundancy. Specifically, sand, silt, and clay are compositional variables whose proportions sum to 100%, leading to perfect collinearity. To reduce redundancy, we removed silt while retaining sand and clay. In addition, LWD was found to be highly correlated with net solar radiation at the daily mean scale (Fig. S1) and was therefore excluded from the final modeling.

In contrast, although the daily mean LST (LST_mean) and air temperature also exhibited strong collinearity, with VIF values exceeding 10 (Fig. S2), we decided to retain both. This decision reflects their physical distinctness and complementary information: LST_mean provides higher spatial resolution (1 km), whereas air temperature offers broader meteorological consistency (9 km). Such differences are particularly important in complex ecosystems such as forests, where canopy structure

and biological processes substantially influence thermal dynamics (Liu et al., 2025).

[Figure]

**Figure S1.** Variance Inflation Factor (VIF) of predictor variables (with LWD)

**Reference**

Liu X., Li Z.-L., Duan S.-B., Leng P., and Si M.: Retrieval of global surface soil and vegetation temperatures based on multisource data fusion, Remote Sens. Environ., 318, 114564, https://doi.org/10.1016/j.rse.2024.114564, 2025.

Peng, S., Ciais, P., Krinner, G., Wang, T., Gouttevin, I., McGuire, A. D., Lawrence, D., Burke, E., Chen, X., Decharme, B., and others: Simulated high-latitude soil thermal dynamics during the past 4 decades, The Cryosphere, 10, 179–192, 2016.

**Reviewer Comment 6:**

*Line 490 "Notably, RMSE at the surface (0 cm) is slightly lower than at 40 cm, possibly due to stronger direct influences from surface cover and meteorological conditions." – This is not the case for Fig. 12 cd. Furthermore, making this statement based on only a few sites is not adequate.*

**Response to Reviewer Comment 6:**

We thank the reviewer for the valuable observation. We agree that the RMSE at 0 cm is not consistently lower than at 40 cm across all stations. Our original statement was overly generalized based on a limited number of sites and may have caused confusion. We have revised the text accordingly to avoid overinterpretation.

**Revised Text (L505-506):**

Site-level accuracy was evaluated using RMSE, which ranged from 0.84 K to 1.80 K

across both depths, indicating strong agreement between predicted and observed values.

**Reviewer Comment 7:**

*Line 535 "Figure. S5 demonstrates that LST is more effective than air temperature in detecting spatial variations in surface Ts in sparsely vegetated areas" – I do not see how this conclusion can be derived from Fig. S5.*

**Response to Reviewer Comment 7:**

We thank the reviewer for this valuable comment. We agree that Fig. S5 alone does not provide direct evidence that LST is more advantageous than air temperature in sparsely vegetated areas. In response, we have revised the text in the manuscript, removed the description related to Fig. S5, and added supporting evidence from Figs. S7 and S8 to more robustly substantiate this conclusion.

**Revised Text (L556-560):**

As shown in Figs. S7 and S8, incorporating LST as an input variable, relative to using only air temperature, significantly enhances overall modeling accuracy and improves performance across sites with different land cover types, with the most pronounced improvements observed in barren land areas.

[Figure]

**Figure S7.** Comparison of Modeling Accuracy with Different Feature Variables (Feature1 represents using both air temperature and LST together with other feature variables, while Feature 2 represents using only air temperature together with other feature variables)

[Figure]

**Figure S8.** Differences in model accuracy across land cover types under different feature variable combinations. (Feature1 represents using both air temperature and LST together with other feature variables, while Feature 2 represents using only air temperature together with other feature variables)

---

## Author Comment (AC3)

**Revisions of Manuscript: ESSD-2025-192**

**Title:** Spatially adaptive estimation of multi-layer soil temperature at a daily time-step across China during 2010-2020

**Author(s):** Xuetong Wang, Liang He, Peng Li, Jiageng Ma, Yu Shi, Qi Tian, Gang Zhao, Jianqiang He, Hao Feng, Hao Shi, Qiang Yu

Dear Reviewer,

We sincerely thank you for your thoughtful comments and constructive suggestions on our manuscript. We have carefully revised the manuscript in response to your feedback, with all changes clearly marked using track changes. In the revised manuscript and accompanying supplementary materials, modifications are highlighted in blue for ease of reference.

Below, we provide a detailed, point-by-point response to each of your comments. For clarity, your original remarks are shown in *italics*, followed by our corresponding replies. We have made every effort to address all concerns comprehensively and to improve the scientific rigor, clarity, and overall quality of the manuscript.

We sincerely appreciate the time and effort you invested in reviewing our work, and we believe the revisions have significantly improved the manuscript.

**Reviewer Comment 1:**

*The method used many data (primarily including in-situ observations and indicator variables) to produce soil temperature. By the way, the in-situ in Figure 3 is wrongly spelled as in-suit. Since these data are with varying spatial scales, and many complicated steps are involved in this procedure to produce the Ts at different depths. I just wonder why the outputted Ts is with that good accuracy. Given than even the acknowledged MODIS LST (nearly Ts at 0 cm) is 1-2K, and it has been taken as an input in this study.*

**Response to Reviewer Comment 1:**

We sincerely thank the reviewer for this valuable comment. The relatively high accuracy of our model can be attributed to the following three aspects:

**1. Complementarity of multi-source information.**

MODIS LST is only one of many predictors and not the dominant determinant. By integrating near-surface air temperature, radiation, precipitation, vegetation indices, topography, and soil texture, the model captures the key drivers of soil thermal dynamics. This multi-source data fusion enables the model to learn complex nonlinear relationships, thereby mitigating the influence of errors from any single predictor (e.g., LST).

**2. Localized modeling based on the rotated quadtree.**

The rotated quadtree adaptively partitions the study domain according to station density, allowing local models to better represent regional heterogeneity. This spatially adaptive approach avoids systematic bias from scale mismatch and significantly improves the model's applicability and stability across diverse regions.

**3. Robust performance under different conditions.**

In subsequent analyses, we compared model performance across seasons and land-use types. Results indicate that model accuracy is relatively higher in spring and autumn than in summer and winter, and is generally greater over croplands, grasslands, and barren lands compared with forests. These patterns further demonstrate that the high accuracy is reasonable and reflects the robustness of the model, rather than an artifact of overfitting.

We also appreciate the reviewer's careful note on the spelling issue in Figure 3. We have corrected "in-suit" to "in-situ" (L342–344 in the revised manuscript).

[Figure]

**Figure 3.** Workflow of the proposed method to obtain multi-layer $T_s$ over the China.

**Reviewer Comment 2:**
*To my knowledge, LST changed very quickly and is seriously affected by cloud. The local observation time differ across China, and most regions in the South are covered by cloud at most time. How you process these data, and whether the accuracy can be guaranteed in your study?*

**Response to Reviewer Comment 2:**
We greatly appreciate the reviewer's attention to this issue. To address the cloud-induced data gaps and temporal mismatch in LST, we implemented the following measures:

**1. Cloud-induced Data Gaps**
Cloud cover, especially in southern China, is indeed a significant challenge. To mitigate this, we reconstructed missing data caused by cloud cover using spatio-temporal interpolation combined with neighboring pixel information. We then applied the Savitzky-Golay smoothing method to generate continuous daily fields, effectively reducing the data gaps caused by cloud interference.

**2. Handling MODIS Daytime and Nighttime LST**
We separately processed the instantaneous daytime and nighttime LST from MODIS, and calculated the mean of these two values to serve as the daily average LST input variable. Compared to instantaneous temperatures, daily mean values are less sensitive to missing data, which helps improve the stability of the data.

**3. Uncertainty in Using LST as an Input Variable**
We acknowledge that using mean_LST as an input variable may introduce some uncertainties, particularly in southern regions where cloud cover leads to more significant data gaps. We have discussed the limitations of this approach and future improvements in the revised discussion section of the manuscript. Despite these uncertainties, considering that mean_LST effectively captures long-term surface temperature trends at a large spatial scale, we decided to use it as a feature for modeling.

We hope these clarifications address the reviewer's concerns regarding cloud effects, temporal mismatches, and the uncertainties introduced by the use of LST as an input variable. The methods we have implemented are well thought out to ensure the accuracy and reliability of the model results.

**Revised Text (L601-L662):**

[revised manuscript text omitted]

**Reviewer Comment 3:**

*On the other hand, the seemingly good accuracy is not that strange. Because the authors used the same ground measurement to validate the estimated values. Although the entire data has been divided into two sections of training and validation. They are actually homologous with the similar schemes by CMA. How about validating the estimated results with data collected from different sources.*

**Response to Reviewer Comment 3:**

We sincerely thank the reviewer for this valuable and necessary comment. In the revised manuscript, we have strengthened the validation design to address this concern by (1) implementing a spatial block cross-validation scheme and (2) incorporating independent validation against flux tower observations, thereby enhancing the independence and credibility of our evaluation.

First, we acknowledge that the CMA operational network is currently the only nationwide source of long-term (≥10 years), large-scale, and multi-layer (0–40 cm) Ts observations in China, and thus forms the most comprehensive basis for constructing a national Ts dataset. To rigorously account for the strong spatial autocorrelation of Ts

and avoid potential data leakage between training and testing subsets, we employed a spatial block cross-validation scheme rather than random splitting. Observation sites were first partitioned into rotated quadtree subregions. Within each subregion, sites were further grouped into spatial blocks by flooring their latitude and longitude values to integer degrees, such that stations sharing the same integer indices (i.e., falling within the same $1° \times 1°$ index) were assigned to the same block. This method ensures that samples within the same spatial block are not simultaneously allocated to both the training and testing subsets, thereby preventing data leakage caused by spatial autocorrelation and providing a more reliable assessment of the model's generalization capability.

Second, to further strengthen independence, we validated the final dataset against daily Ts observations from 18 flux tower sites of the ChinaFLUX network. Measurements at 0, 5, 10, 15, 20, and 40 cm were retained for consistency. Results (Figure 5; Table S2) show that our dataset maintains high accuracy at these independent sites ($R^2 = 0.85$–0.90; RMSE = 3.3–4.2 K), confirming that the accuracy is robust and not merely a product of same-source validation.

Taken together, the validation results from both spatial block cross-validation and independent flux tower observations demonstrate that the spatially adaptive framework we developed achieves strong robustness, reliability, and spatial generalization ability.

**Revised Text (L318-L326):**
To rigorously account for the strong spatial autocorrelation of $T_s$ and avoid potential data leakage between training and testing subsets, we employed a spatial block cross-validation scheme rather than random splitting. Specifically, within each rotated quadtree grid, observation sites were grouped into spatial blocks based on their geographic coordinates: station latitude and longitude were each divided by $1°$ and floored to integer values, and stations sharing the same index were assigned to the same block. This ensured that samples within the same spatial block were not simultaneously assigned to both the training and testing subsets, thereby avoiding data leakage due to spatial autocorrelation and enabling a more reliable evaluation of the model's generalization capability.

Within each spatial grid, the data were partitioned into training (90%) and testing (10%) subsets at the block level. The training subset was further subjected to 10-fold spatial block cross-validation using GridSearchCV to optimize three key hyperparameters: the number of trees (n_estimators), maximum tree depth (max_depth), and learning rate (learning_rate). Detailed parameter settings are provided in Appendix Table S1. The hyperparameter set that yielded the lowest average validation error across the ten folds was selected as optimal. The final model, retrained on the full training set with these parameters, was then evaluated on the held-out testing blocks to assess its generalization ability and examine potential overfitting within each grid.

**Revised Text (L372-L381):**

Furthermore, to enhance the independence of the evaluation, we validated the final dataset against daily $T_s$ observations from 18 flux tower sites of the ChinaFLUX network. For consistency, we retained measurements only at depths of 0, 5, 10, 15, 20, and 40 cm. Metadata for these sites is provided in Table S2, and the corresponding validation results are presented in Figure 5. The evaluation shows that our dataset achieves high accuracy at these independent sites ($R^2 = 0.85$–$0.90$; RMSE $= 3.3$–$4.2$ K), further demonstrating the robustness of our approach. Taken together, the validation results from both spatial block cross-validation and flux tower observations confirm that the spatially adaptive model we developed exhibits reliable accuracy and strong spatial generalization capability.

[Figure]

**Figure 5.** Density scatter plots comparing estimated daily $T_s$ with flux tower observations at different depths

Table.S2 Metadata of daily $T_s$ observations from flux towers used for validation.

| Site | Ecosystem | Depth (cm) | Time series |
|---|---|---|---|
| Baotianman Forest Station | Forest | 0,5,20 | 2010-2014 |
| Changling Rice Paddy Station | Cropland | 5,10,20 | 2018-2020 |
| Daan Cropland Station | Cropland | 0,5,10,15,20 | 2017-2020 |
| Damao Grassland Station | Grassland | 0,5,10,15,20,40 | 2017-2020 |
| Danzhou Rubber Plantation Station | Forest | 5,10,20 | 2010 |
| Haibei Alpine Meadow Station | Grassland | 5,10,15,20,40 | 2015-2020 |
| Haibei Shrubland Station | Grassland | 0,5,20,40 | 2016-2018 |
| Huzhong Boreal Forest Station | Forest | 5,10,20 | 2014-2018 |
| Jinzhou Cropland Station | Cropland | 5,10,15,20,40 | 2011-2014 |
| Lijiang Alpine Meadow Station | Grassland | 5,10,15,20,40 | 2013-2020 |
| Maoershan Forest Station | Forest | 5 | 2016-2018 |
| Panjin Reed Wetland Station | Wetland | 10,20,40 | 2018-2020 |

| | | | |
|---|---|---|---|
| Qianyanzhou Plantation Forest Station | Forest | 5,10,20 | 2011-2015 |
| Ruoergai Alpine Wetland Station | Wetland | 0,5,10,20 | 2013-2020 |
| Sanjiangyuan Alpine Grassland Station | Grassland | 0,5,15 | 2013-2015 |
| Taoyuan Cropland Station | Cropland | 5,10,15,20,40 | 2010-2014 |
| Xishuangbanna Rubber Plantation Station | Forest | 0,5,20 | 2010-2014 |
| Yuanjiang Dry-Hot Valley Savanna Station | Grassland | 5,10,20,40 | 2013-2015 |

**Reviewer Comment 4:**

*The authors used on XGBoost, why not try other machine learning algorithms. It is not sure that XGBoost perform best. Maybe a balance of multiple algorithms is more convincible.*

**Response to Reviewer Comment 4:**

We thank the reviewer for this valuable comment. We agree that other machine learning approaches (e.g., RF, GBDT, LSTM) could in principle be applied to soil temperature estimation. However, the main innovation of our study lies not in algorithm comparison, but in the spatially adaptive modeling framework (rotated quadtree + local modeling + layer-wise cascading), which addresses the challenges posed by spatial non-stationarity and uneven observation distribution in nationwide *Ts* estimation.

We selected XGBoost because it offers clear advantages over alternative methods for large-scale mapping:

**1. Compared to RF**
XGBoost converges faster, is more memory-efficient, and yields lighter prediction models;

**2.Compared to traditional GBDT:**
XGBoost incorporates parallelization, sparse-aware processing, and cache optimization, leading to much higher efficiency on large datasets;

**3. Compared to LSTM and deep learning models:**
XGBoost has lower computational complexity, less dependence on GPUs, and runs efficiently on CPUs, making it more practical for nationwide, daily, decade-long mapping tasks.

Therefore, in the revised manuscript, we emphasized the novelty of the spatially adaptive framework and cited relevant literature to highlight the widespread use of XGBoost in large-scale mapping. The focus of this work is the framework itself rather

than a benchmarking exercise among algorithms. For details, please refer to the revised manuscript.

**Revised Text (L303-L317):**

We adopted the XGBoost (Extreme Gradient Boosting) algorithm as the core regression model for $T_s$ estimation due to its strong predictive performance, computational efficiency, and scalability across large environmental datasets. XGBoost builds an ensemble of regression trees in a stage-wise boosting process, where each tree is trained to minimize the residuals from the previous iteration, leading to a robust and optimized model (Chen and Guestrin, 2016). A key strength of XGBoost is its ability to handle heterogeneous and high-dimensional predictor sets, which are common in geoscience applications involving complex terrain, land cover variability, and climatic gradients. Recent studies have demonstrated its effectiveness in similar domains, including land surface temperature reconstruction (Li et al., 2024), multi-layer soil moisture estimation (Karthikeyan and Mishra, 2021), drought event attribution (Wang et al., 2025), and crop yield prediction (Li et al., 2023). Given these proven strengths and the spatially nonstationary characteristics of $T_s$ in our study area, XGBoost was selected to train localized prediction models within spatial subregions.

In summary, this study not only introduces a new spatially adaptive modeling framework, but also delivers a nationwide $T_s$ dataset that is unique in its resolution, depth coverage, and temporal span. We believe this dataset will provide significant

value for agricultural production, ecosystem modeling, carbon budget assessments, and climate change research, and will serve a broad scientific and applied user community.

---

## Author Response (AR2)

**Revisions of Manuscript:  ESSD-2025-192**

**Title:** Spatially adaptive estimation of multi-layer soil temperature at a daily time-step across China during 2010-2020

**Author(s):** Xuetong Wang, Liang He, Peng Li, Jiageng Ma, Yu Shi, Qi Tian, Gang Zhao, Jianqiang He, Hao Feng, Hao Shi, Qiang Yu

Dear Reviewer,

We sincerely thank you for your thoughtful comments and constructive suggestions on our manuscript. We have carefully revised the manuscript in response to your feedback, with all changes clearly marked using track changes. In the revised manuscript and accompanying supplementary materials, modifications are highlighted in blue for ease of reference.

Below, we provide a detailed, point-by-point response to each of your comments. For clarity, your original remarks are shown in *italics*, followed by our corresponding replies. We have made every effort to address all concerns comprehensively and to improve the scientific rigor, clarity, and overall quality of the manuscript.

We sincerely appreciate the time and effort you invested in reviewing our work.

**Response to Reviewer3_Comments**

**Reviewer Comment 1:**

*The effectiveness of spatial block CV is dependent on the block size being large enough to account for the spatial autocorrelation range of the data. The authors have used 1° (~100 km), which needs to be justified. Some previous studies (e.g., Ploton, et al. 2020 Nature Communications 11 (1): 4540) showed that typical climate variables can exhibit significant spatial correlation up to 500 km. Therefore, the authors should provide a justification for their choice of 1°, for instance by using semivariograms of the observed Ts to determine the distance at which spatial autocorrelation becomes negligible. Otherwise, the data leakage problem is only reduced, not solved.*

**Response to Reviewer Comment 1:**

Thank you very much for your valuable comment. We recognize that the effectiveness of spatial block cross-validation (CV) largely depends on the size of the blocks, which must be sufficiently large to capture the spatial autocorrelation structure inherent in the data. In addition, the spatial block partitioning strategy previously adopted in our study was limited by an inadequately defined threshold and, more importantly, by the fact that the boundaries of adjacent blocks were often very close to each other. As a result, stations located near block edges could still be separated by only short distances and thus remain spatially correlated, making it difficult to fully eliminate potential data leakage under the original partitioning scheme.

Following your suggestion, we performed a semivariogram analysis to determine the optimal distance required to effectively reduce spatial autocorrelation in the $T_s$ data. As shown in Figure S8, the semivariogram of the $T_s$ data reaches a plateau at a distance of approximately 400–500 km, with only minor variations beyond this range. This indicates that spatial autocorrelation in $T_s$ declines substantially around 500 km and becomes negligible at greater distances. This finding is consistent with Ploton et al., (2020), who also reported significant spatial dependence in climate variables over scales of 250~500 km.

Based on these results, we revised our sampling strategy to more effectively mitigate spatial autocorrelation in the $T_s$ station data and to reduce the risk of data leakage between the training and testing subsets arising from spatial dependence. The revised method has been updated in Section 2.3.3 of the manuscript. We believe that these improvements better address the reviewer's concerns regarding spatial autocorrelation and further enhance the reliability of our model's generalization assessment. We sincerely appreciate your insightful comment, which has substantially strengthened the methodological rigor of our study.

**Revised Text in Section 2.3.3 (L354-L375):**
Significant spatial autocorrelation commonly exists among nearby $T_s$ observation sites.

To prevent potential data leakage caused by randomly splitting the training and testing subsets, we conducted the partitioning at the station level and constructed a buffer zone around the selected test station. All other stations located within this buffer were removed, and only stations outside the buffer were retained as the training set. This strategy effectively ensures that samples within the same sub-grid do not appear simultaneously in both the training and testing subsets due to spatial autocorrelation, thereby allowing a more robust and unbiased assessment of the model's generalization performance.

Specifically, considering the availability of sufficient training samples, one station was randomly selected as the test sample within each sub-grid. A 500 km buffer was subsequently created around the test station, with the radius determined based on the effective distance for reducing spatial autocorrelation among stations as shown in Appendix Figure S8. All stations within the buffer were excluded, and only those outside the buffer were used for model training. Subsequently, five-fold cross-validation was performed at the station level, and GridSearchCV was used to optimize three key hyperparameters: the number of trees (n_estimators), maximum tree depth (max_depth), and learning rate (learning_rate). The search ranges for these parameters are provided in Appendix Table S1. The optimal hyperparameter combination was identified by minimizing the mean validation error. Finally, the model was retrained on the full training subset using the optimized parameters and evaluated on the spatially independent test sample to rigorously assess its generalization capability.

**Here are the revisions, supplemented in the Appendix (L95-L100):**

[Figure]

**Figure S8.** Experimental and theoretical semivariograms of annual mean $T_s$ at 0 cm (The spherical, exponential, and Gaussian models are fitted for comparison).

**Reference**

Ploton, P., Mortier, F., Réjou-Méchain, M., Barbier, N., Picard, N., Rossi, V., Dormann, C., Cornu, G., Viennois, G., Bayol, N., Lyapustin, A., Gourlet-Fleury, S., and Pélissier, R.: Spatial validation reveals poor predictive performance of large-scale ecological mapping models, Nat. Commun., 11, 4540, https://doi.org/10.1038/s41467-020-18321-y, 2020.

**Reviewer Comment 2:**

*In their response, the authors present a new figure (labeled Figure 1 in the response letter) that validates the model's ability to capture the spatial distribution of annual mean Ts. However, the number of points in this figure looks far greater than the ~200 sites that would constitute a 10% test set, suggesting that all stations are included in this validation. For a validation to be a true test of generalization, it must be performed exclusively on the held-out test blocks.*

*Furthermore, I suggest an enhancement: color-code the points in the plot by their parent quadtree grid. This would provide a visual assessment of the performance of the different localized models.*

**Response to Reviewer Comment 2:**

Thank you for your valuable suggestions, which have helped us further enhance the rigor and clarity of our spatial validation and visualization. In the previous response letter, Figure 1 indeed included all stations rather than only the independent test-block stations. Therefore, in the revised manuscript, we validate the model's ability to capture the spatial distribution of annual mean $T_s$ using only the test set.

Regarding your suggestion to color-code the points based on their quadtree grids, our quadtree system is rotated at six different angles. Under different rotations, both the number of sub-grids and their identifiers vary, and the selected test set is not fixed across rotations. As a result, it is not feasible to apply a consistent color-coding scheme to represent grid membership within a single figure.

Nevertheless, the revised Figure S12 clearly demonstrates that across all rotation angles and corresponding grid structures, the localized models consistently and accurately capture the spatial distribution of annual mean $T_s$ when evaluated on the independent test set. This further confirms the robustness of the rotated-quadtree modeling framework.

**Here are the revisions, supplemented in the Appendix (L118-L121):**

[Figure]

**Figure S12.** Comparison between estimated and observed annual mean $T_s$ across six depths (0~40 cm)

**Reviewer Comment 3:**

*The addition of an independent validation using 18 flux tower sites is a great improvement. The authors present the daily time-series comparison in the new Figure 5. To maintain consistency with the most robust validation practices, I suggest that the authors also present the annual mean Ts (spatial-only validation) comparison between model and observation.*

**Response to Reviewer Comment 3:**

We appreciate the reviewer's insightful suggestion. Following your recommendation, we have now included an additional spatial-only validation based on the annual mean $T_s$ derived from the 18 independent flux tower sites. This new analysis directly assesses the model's ability to reproduce the spatial distribution of annual mean $T_s$ without relying on temporal information.

The results are presented in the newly added Figure S13, which compares the observed and estimated annual mean $T_s$ across six soil depths (0~40 cm). Despite the small number of independent sites and the strong climatic heterogeneity among tower locations, the model achieves reasonable agreement with observations, demonstrating its capacity to capture the spatial variability of annual mean $T_s$ under independent conditions. This addition ensures consistency with best practices in model evaluation and provides a more comprehensive and rigorous validation of the model's spatial performance.

**Here are the revisions, supplemented in the Appendix (L122-L126):**

[Figure]

Figure S13. Comparison between estimated and FLUX towers annual mean $T_s$ across six depths (0~40 cm)

**Response to Reviewer4_Comments**

**Specific comments:**

**Reviewer Comment 1:**

*Discussion on time-series is missing. How has the Ts changed during this decade?*

**Response to Reviewer Comment 1:**

Thank you for this helpful comment. We agree that our current manuscript does not include a full discussion of long-term $T_s$ changes over the entire decade (2010–2020). In our study, interannual variability of $T_s$ during this period is relatively small, and our primary objective was to evaluate the model's ability to reproduce $T_s$ dynamics rather than to analyze long-term climate trends. Therefore, instead of presenting a full decadal trend analysis, we selected four representative stations and examined in detail their daily variations of air temperature, LST and $T_s$ for the period 2018–2019 as illustrative examples.

**Revised Text (L524-L541):**

To further assess the temporal performance of $T_s$ estimation, Fig. 11 presents the time series of estimated $T_s$ alongside in-situ measurements at four randomly selected stations (e.g., Station 56748, 99.18°E, 25.12°N) from January 2018 to January 2020. The figure displays $T_s$ at two depths (0 cm and 40 cm), including estimated $T_s$ (Estimated_0cm, Estimated_40cm), in-situ $T_s$ (In-situ_0cm, In-situ_40cm), daily mean land surface temperature (Daily_mean_LST), and 2-meter air temperature (Temperature_2m). The air temperature shows distinct seasonal cycles, while $T_s$ exhibits smoother temporal variations. In general, $T_s$ reaches its peak during summer and its minimum in winter, though its temporal dynamics vary with soil depth. Specifically, $T_s$ at 0 cm responds rapidly to air temperature changes and exhibits larger amplitude variations, while $T_s$ at 40 cm shows slower responses and a noticeable lag, reflecting the damping effect of vertical heat conduction. Site-level accuracy was evaluated using RMSE, which ranged from 1.24 K to 2.05 K across both depths, indicating strong agreement between predicted and observed values. Overall, the time series analysis confirms the robustness and reliability of the model in estimating $T_s$ across varying depths, offering valuable insights into regional soil thermal dynamics.

[Figure]

**Figure 11.** Time series of the Estimated_0cm, Estimated_40cm, Daily_mean_LST, and Temperature_2m at four sites from different regions between 2018-2019.

**Reviewer Comment 2:**

*Too many figures in the main text. Suggest to move some into supplemental material.*

**Response to Reviewer Comment 2:**

Thank you for your helpful suggestion. Following your recommendation, we have streamlined the presentation of figures in the main text. Specifically, we have merged the original Sections 3.1 and 3.2 into a new Section 3.1 to reduce redundancy and improve the clarity of the Results. In addition, Figure 7 from the former Section 3.2 has been moved to the Supplementary Material. These adjustments help keep the main manuscript focused on essential results while ensuring that all supporting visualizations remain accessible in the appendix.

**Reviewer Comment 3:**

*Adaptive scaling can lead to confusion when interpreting the spatial resolution of a dataset. I would recommend authors to report the "effective spatial resolution" information of each grid, or store it as a separated supporting dataset. Otherwise, it is hard to be used for comparison against other regional/global products with fixed spatial resolution.*

**Response to Reviewer Comment 3:**

Thank you very much for your thoughtful comment. We fully understand the concern that adaptive spatial partitioning may cause confusion regarding the spatial resolution of the final dataset. Here we would like to clarify that, although our modeling framework uses a rotated-quadtree structure to adaptively partition the training space

based on the density of in-situ observations, the final $T_s$ product is always generated at a fixed spatial resolution of 1 km.

The adaptive quadtrees are used only during the modeling stage to train localized XGBoost models under multiple rotation angles. After model training, the predictions from all rotated-quadtree configurations are aggregated and mapped onto a uniform 1-km grid across China. Therefore, the effective spatial resolution of the final product does not vary across space, and all output grids represent the same 1-km resolution regardless of quadtree size during training.

We hope this clarification addresses your concern, and we appreciate your attention to dataset usability and transparency.

**Reviewer Comment 4:**
*The dataset is claimed to be freely accessible, but I cannot access the data through FTP. Authors need to verify and check the link, or upload a copy to an open access repository, otherwise it does not meet the requirement of ESSD.*
**Response to Reviewer Comment 4:**
Thank you for pointing out this issue. We sincerely apologize for the inconvenience caused by the temporary inaccessibility of the FTP link. After receiving your comment, we immediately contacted the data center and confirmed that the FTP remote server itself had not experienced any malfunction. To ensure that the dataset fully meets ESSD's data accessibility requirements, we have taken the following steps:
(1) The complete $T_s$ dataset has been re-uploaded to the National Tibetan Plateau Data Center (TPDC), and the DOI download links have been fully verified and are functioning properly. Two stable access links are now provided for users:
https://doi.org/10.11888/Terre.tpdc.302333
https://cstr.cn/18406.11.Terre.tpdc.302333
(2) If access issues persist, we kindly recommend users to switch to another network environment (e.g., non-campus network or networks without firewall restrictions), as some institutional networks may block external data repository access.
At present, the dataset is fully accessible through the verified DOI links and fully satisfies the data availability requirements of ESSD. We sincerely appreciate your valuable feedback.

**Technical corrections:**

**Reviewer Comment 1:**
*Line 108 - 116: This part can be moved to objectives. When describing each objective, state the uniqueness/improvement from your study.*
**Response to Reviewer Comment 1:**
Thank you very much for your valuable suggestion. Following your recommendation, we have revised the corresponding part of the Introduction. The updated content has now been incorporated into the revised manuscript.

**Revised Text (L105-L135):**

Recent advances in spatially adaptive modeling have increasingly emphasized the importance of addressing spatial heterogeneity and uneven sampling density in environmental datasets. Classical quadtree structures and related hierarchical spatial data models provide the theoretical foundation for constructing adaptive, variable-sized spatial partitions, enabling efficient organization of multiscale spatial information through recursive subdivision (Samet, 1984). Building on this foundation, Lagonigro et al., (2020) developed the AQuadtree R package, which provides an adaptive spatial partitioning framework capable of generating variable-sized grid cells according to the spatial distribution of observations. This adaptive partitioning produces finer grids in data-dense regions and coarser grids where observations are sparse, ensuring a spatial structure that better reflects sampling heterogeneity and improves the model's capacity to capture localized spatial variability. Extending this idea, we develop a rotated-quadtree strategy that applies multiple orientation angles during the quadtree subdivision process. This enhancement allows the model to capture spatial heterogeneity from multiple directional perspectives, and averaging predictions across rotation angles substantially reduces the boundary artifacts that may arise from single-angle grid partitioning, ultimately improving the robustness of local modeling under complex environmental gradients.

To address the irregular station distribution, and non-stationarity commonly encountered in large-scale $T_s$ estimation, we construct a spatially adaptive modeling framework based on the rotated quadtree approach. Within each grid cell, multi-source environmental predictors are integrated with in situ station records, and $T_s$ is estimated using XGBoost models. Based on this framework, the objectives of this study are to: (1) construct a spatially adaptive modeling system; (2) generate a multi-layer $T_s$ dataset at a daily time-step and one kilometer resolution in China from 2010-2020; and (3) evaluate the dataset through independent validation with flux tower observations and benchmarking against widely used $T_s$ products. The proposed methodology could directly address the scaling challenges induced by spatial heterogeneity and uneven data distribution. The generated products would provide a robust foundation for high-resolution environmental modeling, precision agriculture and climate impact assessments.

**Reference**

Lagonigro, R., Oller, R., Martori, J.C., 2020. AQuadtree: An R package for quadtree anonymization of point data.

Samet, H., 1984. The quadtree and related hierarchical data structures. ACM Comput. Surv. CSUR 16, 187–260.

**Reviewer Comment 2:**

*Line 110: I would be curious about how to use your data product, since most of the observed/modeled data products have uniform spatial resolution, and your varying resolution product will be hard for intercomparison against them.*

**Response to Reviewer Comment 2:**
Thank you for raising this important question. We would like to clarify that although our modeling framework employs an adaptive, rotated-quadtree structure during the training stage, the final $T_s$ product is always generated on a uniform 1-km grid across China. The adaptive partitioning only determines how local XGBoost models are trained according to site density, and it does not affect the spatial resolution of the final gridded product. To avoid potential misunderstanding, we provide the following clarification here:
(1) The adaptive grid system is used solely for local model training;
(2) The final predicted $T_s$ fields are mapped to a fixed 1-km grid;
Thus, users can apply our $T_s$ dataset in the same manner as any other 1-km gridded environmental product, without concerns related to heterogeneous spatial resolution.

**Reviewer Comment 3:**
*Line 252: How did you treat auxiliary predictors differently from main predictors?*
**Response to Reviewer Comment 3:**
Thank you for raising this point. In addition to the distinction between main and auxiliary predictors, we would like to further clarify how multicollinearity was handled and why both air temperature and LST were retained in the final model.

Before modeling, we conducted a comprehensive variance inflation factor (VIF) analysis for all predictor variables to remove those exhibiting strong multicollinearity. The results are shown in the Supplementary Figures. As expected, air temperature and satellite-derived LST showed high collinearity. To determine whether both variables should be retained, we performed a comparative modeling experiment using two predictor combinations:
(1) Air temperature + other predictors
(2) Air temperature + LST + other predictors
As shown in Supplementary Figure S3 and Figure S4, the second combination (air temperature + LST + other predictors) consistently produced the best predictive performance across depths. This indicates that although air temperature and LST are correlated, they contain complementary thermal information—air temperature captures large-scale atmospheric conditions, whereas LST provides fine-resolution surface radiometric temperature signals.

Therefore, despite their statistical collinearity, both variables were retained in the final model to maximize predictive accuracy. This rationale has been added to Section 2.3.1 of the revised manuscript.

**Here are the revisions, supplemented in the Appendix (L5-L20):**

[Figure]

**Figure S3.** Comparison of Modeling Accuracy with Different Feature Variables (Feature1 represents using both air temperature and LST together with other feature variables, while Feature 2 represents using only air temperature together with other feature variables)

[Figure]

**Figure S4.** Differences in model accuracy across land cover types under different feature variable combinations. (Feature1 represents using both air temperature and LST together with other feature variables, while Feature 2 represents using only air temperature together with other feature variables)

**Reviewer Comment 4:**

*Line 268: Change "fig. 3" to "fig. 2"*

**Response to Reviewer Comment 4:**

Thank you for your correction. We have updated the figure citation accordingly in the revised manuscript.

**Reviewer Comment 5:**

*Line 300: "Such differences are particularly important in complex ecosystems such as forests, where canopy structure and biological processes substantially influence thermal dynamics (Liu et al., 2025)." I'm not quite understanding this sentence and its connection to the context, can you explain a bit or rephrase?*

**Response to Reviewer Comment 5:**

Thank you for pointing out the lack of clarity in this sentence. We agree that the original description did not clearly convey its intended meaning nor its connection to the context. The purpose of this sentence was to justify why both daily mean LST and air temperature were retained despite their high VIF values, by emphasizing their physical differences and complementary thermal information. To improve clarity, we have rewritten the sentence in the revised manuscript.

**Revised Text (L285-L297):**

Although the daily mean LST (LST_mean) and air temperature exhibit high collinearity (VIF > 10; Fig. S2), we chose to retain both variables because they represent different thermal information. LST_mean captures high-resolution surface radiative temperature signals, whereas air temperature reflects broader-scale atmospheric thermal conditions. In ecosystems with complex canopy structures, such as forests, the canopy can alter radiative transfer processes and cause LST to deviate from the true subsurface thermal environment(Liu et al., 2025). Therefore, the two variables provide complementary thermal information that helps better characterize soil thermal dynamics.In addition, we compared the model performance under different combinations of predictor variables (Fig. S3 and Fig. S4). The results show that the combination of air temperature + LST + other predictors achieved the best modeling accuracy at the surface soil layers. Therefore, retaining both air temperature and LST in the final model is reasonable and necessary.

**Reference**

Liu X., Li Z.-L., Duan S.-B., Leng P., Si M., 2025. Retrieval of global surface soil and vegetation temperatures based on multisource data fusion. Remote Sens. Environ. 318, 114564. https://doi.org/10.1016/j.rse.2024.114564

**Reviewer Comment 6:**

*Line 440: Should MAE equation be the difference between prediction and observations divided by total sample number? Is "2" a typo?*

**Response to Reviewer Comment 6:**

Thank you for pointing this out. You are correct — the denominator should represent the total number of samples ($N$). The "2" in the MAE equation was a typographical

error. The correct formulation of the Mean Absolute Error (MAE) is:

$$MAE = \frac{\sum_{i=1}^{N} |x_i - y_i|}{N} \qquad (1.1)$$

where $x_i$ and $y_i$ denote the observed and predicted values, respectively. We have corrected this in the revised manuscript.

**Reviewer Comment 7:**
*Fig. 6, 7: There are narrow regions with quite dark color over the eastern coast. Are these regions with low R-square and RMSE or just the boundary? Please clarify.*
**Response to Reviewer Comment 7:**
Thank you for your observation. The narrow dark regions in Figures 6 and 7 along the eastern coast correspond to islands and boundary areas, rather than regions with low R² or high RMSE values. We apologize for any confusion this may have caused.

**Reviewer Comment 8:**
*Line 579: Did you upscale your fine resolution results from XGBoost before comparing it to other data products?*
**Response to Reviewer Comment 8:**
Thank you for your question. Before comparing with the ERA5_Land and GLDAS products, all three datasets were resampled to match the spatial resolution of ERA5_Land. This ensured consistency in spatial scale and allowed for a fair and accurate evaluation.

**Reviewer Comment 9:**
*Section 3.1 and 3.2 both are evaluated at site level, so I suggest merging them.*
**Response to Reviewer Comment 9:**
As you correctly pointed out, both Sections 3.1 and 3.2 focused on model evaluation at the site level. Accordingly, we have merged the original Sections 3.1 and 3.2 into a single new Section 3.1 based on your suggestion. In addition, Figure 7 from the former Section 3.2 has been moved to the Appendix.

**Reviewer Comment 10:**
*section 3.3: Fine resolution product shall be able to reflect the response of Ts to elevation. I did not see any discussion on this point.*
**Response to Reviewer Comment 10:**
Thank you for this valuable comment. In the revised manuscript, we have expanded our discussion on how the fine-resolution $T_s$ product captures the response of $T_s$ to elevation. In this section, we compare our product with other existing datasets and highlight the advantages of our 1-km $T_s$ estimates in representing spatial variations across different topographic conditions. Please refer to Section 3.3 of the revised manuscript for the detailed discussion.

**Revised Text (L470-L479):**

Figure 9 presents a comparison of the $T_s$ products at the 0 cm depth with the ERA5-Land and GLDAS 2.1 reanalysis datasets, including both national-scale patterns (Fig. 9a–c) and zoomed-in regional details (Fig. 9d–f). Compared with the two reanalysis products, our generated $T_s$ dataset exhibits substantially finer spatial resolution, enabling a clearer representation of localized spatial heterogeneity. As illustrated in the zoomed-in panels of Figure 9, our $T_s$ product accurately captures terrain- and elevation-driven temperature gradients in regions with strong topographic variability, such as the transition zone from the Sichuan Basin to the margins of the QTP. In contrast, the coarse spatial resolution of ERA5-Land and GLDAS 2.1 tends to smooth out these fine-scale topographic effects, resulting in a loss of spatial detail.

**Reviewer Comment 11:**

*Fig. 12: Authors shall improve the figure quality. Lines are too thin and resolution seems to be low.*

**Response to Reviewer Comment 11:**

Thank you for your helpful comment regarding the figure quality. We have improved the visualization by increasing the line thickness and rendering resolution. The figure has been regenerated at higher quality and the updated version has been incorporated into the revised manuscript. The original Figure 12 has now been updated and is presented as Figure 11 in the revised manuscript.

[Figure]

**Figure 11.** Time series of the Estimated_0cm, Estimated_40cm, Daily_mean_LST, and Temperature_2m at four sites from different regions between 2018-2019.

**Reviewer Comment 12:**

*Line 822: Missing soil moisture can also bring substantial error in capturing daily or sub-daily variations of soil temperature. Authors shall discuss it as well.*

**Response to Reviewer Comment 12:**

Thank you for this insightful comment. We agree that soil moisture plays an important role in regulating daily and sub-daily soil temperature dynamics, and the absence of soil moisture information can introduce additional uncertainty in $T_s$ estimation. In the revised manuscript, we have added a dedicated discussion on this issue, highlighting how the lack of soil moisture data may affect the model's performance and the potential pathways for future improvements. The detailed discussion can be found in Section 4.3.

**Revised Text (L674-L682):**

Short-term changes in soil moisture alter fundamental soil thermal properties, including heat capacity, thermal conductivity, and thermal diffusivity, which in turn control heat transfer processes and sub-daily $T_s$ dynamics. (Abu-Hamdeh, 2003; Subin et al., 2013). Consequently, the absence of soil moisture information may introduce additional uncertainty when modeling daily and sub-daily $T_s$ dynamics, especially at deeper layers. Incorporating high-resolution soil moisture datasets in future work would improve the representation of soil hydrothermal interactions and further enhance $T_s$ estimation accuracy.

**Reference**

Abu-Hamdeh, N.H., 2003. Thermal properties of soils as affected by density and water content. Biosyst. Eng. 86, 97–102.

Subin, Z.M., Koven, C.D., Riley, W.J., Torn, M.S., Lawrence, D.M., Swenson, S.C., 2013. Effects of soil moisture on the responses of soil temperatures to climate change in cold regions. J. Clim. 26, 3139–3158.

**Reviewer Comment 13:**

*Line 826: "explained by the physical characteristics of soil temperature". Should this be "the physical characteristics of soil texture profile"?*

**Response to Reviewer Comment 13:**

Thank you for this helpful clarification. We agree that the original expression was imprecise. The depth-dependent behavior of $T_s$ is indeed more closely related to the physical characteristics of the soil profile, including soil texture, bulk density, and thermal properties, rather than to the "characteristics of soil temperature" itself. We have revised the manuscript accordingly to replace the phrase with "the physical characteristics of the soil profile." This improves both accuracy and clarity.

**Revised Text (L663-L664):**

This depth-dependent pattern can be explained by the physical characteristics of the soil profile.

**Reviewer Comment 14:**

*Line 830: "dampen high-frequency fluctuations and stabilize the relationship between predictors and Ts". This conclusion is a bit misleading. In relatively deeper soil depth, "dampen high-frequency fluctuations" is true, but this does not stabilize the predictor -*

$T_s$ relationship, but reduces the signal of the predictor. Following these reasons, "thereby improving performance at these depths" is not true. The lower RMSE/R2 is a consequence of low variability of deeper soil temperature itself, not reflecting a better performance for deeper soil temperature. Please revise.

**Response to Reviewer Comment 14:**

We appreciate this insightful comment. We agree that the original wording may have been misleading. The reduced RMSE and increased R² at intermediate depths do not necessarily imply that the predictor–$T_s$ relationships are stronger at those depths. Instead, the improved metrics primarily reflect the dampened variability of $T_s$ caused by thermal buffering and increased heat capacity in mid-soil layers. In the revised manuscript, we have rewritten the sentence to clarify that the performance metrics at middle depths are largely a consequence of reduced temporal and spatial fluctuations in $T_s$, rather than inherently better model performance or stronger predictor–response relationships.

**Revised Text (L667-L670):**

In contrast, intermediate soil layers buffer high-frequency temperature fluctuations through thermal diffusion and higher heat capacity. As a result, $T_s$ becomes more stable with lower natural variability at these depths, leading to lower RMSE and higher R² values.

**Reviewer Comment 15:**

Line 832: "At greater depths, however, surface-level errors propagate downward through the cascading framework, resulting in reduced accuracy—particularly during summer and winter." Can this be a consequence from uncertain soil texture profile input? As soil goes deeper, the uncertainty from soil texture will accumulate and becomes higher,

**Response to Reviewer Comment 15:**

Thank you for this valuable suggestion. We agree that uncertainty in soil texture inputs can accumulate with depth and may contribute to the reduced accuracy observed at deeper soil layers. Although our modeling framework incorporates multi-layer soil texture information, uncertainties in deep soil texture may propagate through the cascading prediction structure. We have added this point to Section 4.3 of the Discussion, acknowledging that soil texture uncertainty is an additional factor influencing deep-layer $T_s$ errors.

**Revised Text (L671-L674):**

At deeper layers, prediction accuracy decreases because surface-level errors propagate downward through the hierarchical modeling framework, and uncertainties in soil texture inputs gradually accumulate with depth; during periods such as summer and winter, these combined uncertainties may be further amplified.

**Reviewer Comment 16:**

Line 836: Another reason is that the LST product reflects the temperature that the remote sensor observed. So it can be either soil surface temperature, snow temperature

*or temperature at canopy top. Please address how this uncertainty source can impact your results.*

**Response to Reviewer Comment 16:**

We greatly appreciate this important comment. We agree that the LST product represents the radiometric temperature observed by the satellite sensor, and the specific temperature it reflects may vary depending on land cover and seasonal conditions. As a result, LST may correspond to surface $T_s$, snow surface temperature, or canopy-top temperature. This inherent ambiguity indeed introduces an additional source of uncertainty in the estimation of $T_s$. We have added a discussion of this issue in Section 4.3 of the revised manuscript.

**Revised Text (L683-L703):**

Seasonal variations and differences in land cover also contribute to the spatiotemporal differences in model performance. As shown in Figures 7 and 8, the model performs better in spring and autumn, whereas its accuracy declines in summer and winter. In summer, vigorous vegetation growth and canopy closure alter surface–atmosphere energy exchange processes and weaken the relationship between canopy temperature and subsurface $T_s$, thereby reducing the effectiveness of LST as a proxy for near-surface $T_s$ (Kropp et al., 2020; Cui et al., 2022). Moreover, because satellite sensors measure radiometric temperature, LST in densely vegetated regions often represents canopy-top temperature rather than the surface $T_s$, introducing an additional source of uncertainty. In winter, snow cover further increases complexity: the high albedo of snow reduces net radiation (Loranty et al., 2014; Li et al., 2018), and its insulating effect weakens the soil's response to cold-air fluctuations (Zhang, 2005; Myers-Smith et al., 2015). Meanwhile, Meanwhile, freezing of soil water alters soil thermal conductivity and heat capacity, and frequent freeze–thaw cycles introduce nonlinear dynamics into $T_s$, increasing modeling uncertainty (Li et al., 2023a; Imanian et al., 2024). Although our multi-source adaptive modeling framework demonstrates robust performance across varying depths and environmental conditions, it does not explicitly represent the physical mechanisms governing vertical heat transfer. Future research could incorporate deep learning models capable of learning complex spatiotemporal dependencies to enhance the physical interpretability of $T_s$ variations across time, space, and depth.

**Reference**

Cui, X., Xu, G., He, X., Luo, D., 2022. Influences of seasonal soil moisture and temperature on vegetation phenology in the Qilian Mountains. Remote Sens. 14, 3645. https://doi.org/10.3390/rs14153645

Imanian, H., Mohammadian, A., Farhangmehr, V., Payeur, P., Goodarzi, D., Hiedra Cobo, J., Shirkhani, H., 2024. A comparative analysis of deep learning models for soil temperature prediction in cold climates. Theor. Appl. Climatol. 155, 2571–2587. https://doi.org/10.1007/s00704-023-04781-x

Kropp, H., Loranty, M.M., Natali, S.M., Kholodov, A.L., Rocha, A.V., Myers-Smith, I., Abbot, B.W., Abermann, J., Blanc-Betes, E., Blok, D., Blume-Werry, G., Boike,

J., Breen, A.L., Cahoon, S.M.P., Christiansen, C.T., Douglas, T.A., Epstein, H.E., Frost, G.V., Goeckede, M., Høye, T.T., Mamet, S.D., O'Donnell, J.A., Olefeldt, D., Phoenix, G.K., Salmon, V.G., Sannel, A.B.K., Smith, S.L., Sonnentag, O., Vaughn, L.S., Williams, M., Elberling, B., Gough, L., Hjort, J., Lafleur, P.M., Euskirchen, E.S., Heijmans, M.M., Humphreys, E.R., Iwata, H., Jones, B.M., Jorgenson, M.T., Grünberg, I., Kim, Y., Laundre, J., Mauritz, M., Michelsen, A., Schaepman-Strub, G., Tape, K.D., Ueyama, M., Lee, B.-Y., Langley, K., Lund, M., 2020. Shallow soils are warmer under trees and tall shrubs across arctic and boreal ecosystems. Environ. Res. Lett. 16, 015001. https://doi.org/10.1088/1748-9326/abc994

Li, Q., Ma, M., Wu, X., Yang, H., 2018. Snow cover and vegetation-induced decrease in global albedo from 2002 to 2016. J. Geophys. Res. Atmospheres 123, 124–138. https://doi.org/10.1002/2017JD027010

Li, X., Zhu, Y., Li, Q., Zhao, H., Zhu, J., Zhang, C., 2023. Interpretable spatio-temporal modeling for soil temperature prediction. Front. For. Glob. Change 6, 1295731. https://doi.org/10.3389/ffgc.2023.1295731

Loranty, M.M., Berner, L.T., Goetz, S.J., Jin, Y., Randerson, J.T., 2014. Vegetation controls on northern high latitude snow-albedo feedback: Observations and CMIP 5 model simulations. Glob. Change Biol. 20, 594–606. https://doi.org/10.1111/gcb.12391

Myers-Smith, I.H., Elmendorf, S.C., Beck, P.S.A., Wilmking, M., Hallinger, M., Blok, D., Tape, K.D., Rayback, S.A., Macias-Fauria, M., Forbes, B.C., Speed, J.D.M., Boulanger-Lapointe, N., Rixen, C., Lévesque, E., Schmidt, N.M., Baittinger, C., Trant, A.J., Hermanutz, L., Collier, L.S., Dawes, M.A., Lantz, T.C., Weijers, S., Jørgensen, R.H., Buchwal, A., Buras, A., Naito, A.T., Ravolainen, V., Schaepman-Strub, G., Wheeler, J.A., Wipf, S., Guay, K.C., Hik, D.S., Vellend, M., 2015. Climate sensitivity of shrub growth across the tundra biome. Nat. Clim. Change 5, 887–891. https://doi.org/10.1038/NCLIMATE2697

Zhang, T., 2005. Influence of the seasonal snow cover on the ground thermal regime: An overview. Rev. Geophys. 43. https://doi.org/10.1029/2004RG000157

**Response to Reviewer5_Comments**

**Reviewer Comment 1:**

*The Introduction would benefit from incorporating recent advances in spatial adaptive modeling using quadtree recursive retrieval, with explicit discussion of the comparative advantages of the proposed methodology.*

**Response to Reviewer Comment 1:**

Thank you very much for this constructive suggestion. We agree that a more systematic discussion of recent advances in spatial adaptive modeling based on quadtree recursive partitioning would strengthen the Introduction. In the revised manuscript, we have expanded the Introduction to incorporate relevant developments in quadtree-based spatial modeling and have further clarified the advantages of our proposed rotated-quadtree framework.

**Revised Text (L105-L135):**

Recent advances in spatially adaptive modeling have increasingly emphasized the importance of addressing spatial heterogeneity and uneven sampling density in environmental datasets. Classical quadtree structures and related hierarchical spatial data models provide the theoretical foundation for constructing adaptive, variable-sized spatial partitions, enabling efficient organization of multiscale spatial information through recursive subdivision (Samet, 1984). Building on this foundation, Lagonigro et al., (2020) developed the AQuadtree R package, which provides an adaptive spatial partitioning framework capable of generating variable-sized grid cells according to the spatial distribution of observations. This adaptive partitioning produces finer grids in data-dense regions and coarser grids where observations are sparse, ensuring a spatial structure that better reflects sampling heterogeneity and improves the model's capacity to capture localized spatial variability. Extending this idea, we develop a rotated-quadtree strategy that applies multiple orientation angles during the quadtree subdivision process. This enhancement allows the model to capture spatial heterogeneity from multiple directional perspectives, and averaging predictions across rotation angles substantially reduces the boundary artifacts that may arise from single-angle grid partitioning, ultimately improving the robustness of local modeling under complex environmental gradients.

To address the irregular station distribution, and non-stationarity commonly encountered in large-scale $T_s$ estimation, we construct a spatially adaptive modeling framework based on the rotated quadtree approach. Within each grid cell, multi-source environmental predictors are integrated with in situ station records, and $T_s$ is estimated using XGBoost models. Based on this framework, the objectives of this study are to: (1) construct a spatially adaptive modeling system; (2) generate a multi-layer $T_s$ dataset at a daily time-step and one kilometer resolution in China from 2010-2020; and (3) evaluate the dataset through independent validation with flux tower observations and benchmarking against widely used $T_s$ products. The proposed methodology could directly address the scaling challenges induced by spatial heterogeneity and uneven

data distribution. The generated products would provide a robust foundation for high-resolution environmental modeling, precision agriculture and climate impact assessments.

**Reference**

Lagonigro, R., Oller, R., Martori, J.C., 2020. AQuadtree: An R package for quadtree anonymization of point data.

Samet, H., 1984. The quadtree and related hierarchical data structures. ACM Comput. Surv. CSUR 16, 187–260.

**Reviewer Comment 2:**

*Additional clarification regarding auxiliary variable selection criteria is warranted. Beyond literature review, principles such as correlation analysis should be elaborated. It is recommended to first validate the existence of spatially non-stationary relationships using bivariate local Moran's I, then proceed with spatial partitioning and predictive modeling via quadtree recursive retrieval.*

**Response to Reviewer Comment 2:**

Thank you very much for your valuable suggestion. We agree that it is necessary to further clarify the criteria for selecting auxiliary variables and the evaluation of spatial non-stationarity. In the revised manuscript, we focus on the local form of spatial association analysis and provide a more detailed explanation of the analytical procedures used in this study.

**Here are the revisions, supplemented in the Appendix (L23-L50):**

To examine whether the relationships between $T_s$ (GST_Avg) and the auxiliary variables exhibit spatial non-stationarity, we employed the Local Bivariate Moran's I, a local statistic within the Local Indicators of Spatial Association (LISA) framework. This method allows us to reveal localized spatial associations and spatially varying relationships between the target variable (X) and the spatially lagged auxiliary variable (Wy). First, we constructed a spatial weights matrix using the K-nearest neighbors method (K = 8). This configuration is suitable for the irregular spatial distribution of meteorological stations across China and ensures that each station has a comparable number of spatial neighbors.

Based on this spatial weights structure, we calculated the Local Bivariate Moran's I between GST_Avg (X) and elevation (Y), and obtained permutation-based p-values. We then computed the spatially lagged auxiliary variable (Wy) and classified each station into one of four significant LISA cluster types ($p < 0.05$): High–High (red), High–Low (green), Low–High (purple), and Low–Low (blue). Stations with non-significant local associations ($p \geq 0.05$) are shown in gray. As illustrated in Figure S5, approximately 64% of the stations exhibit statistically significant local spatial associations, and all four cluster types occur across different regions of China. These spatially heterogeneous local association patterns clearly indicate pronounced spatial non-stationarity in the $T_s$–elevation relationship.

These findings further demonstrate the necessity of adopting a spatially adaptive modeling framework. Accordingly, the rotated quadtree model developed in this study is well justified, as it can effectively capture localized variations in predictor–response relationships.

[Figure]

**Figure S5.** Spatial patterns of the bivariate Local Moran's I between GST_Avg and elevation at meteorological stations across China.

**Revised Text (L300-L305):**

We applied the Local Bivariate Moran's I analysis to assess the local spatial relationship between surface $T_s$ (GST_Avg) and elevation as an illustrative example (Fig. S5). The results reveal significant spatial variations in their local association ($p < 0.05$), indicating pronounced spatial non-stationarity in the $T_s$–elevation relationship. These findings justify the need for a spatially adaptive modeling strategy capable of capturing localized heterogeneity.

**Reviewer Comment 3:**

*Methodological details concerning quadtree rotation require elaboration: Can this approach achieve complete coverage of the study area? Is 360-degree rotation necessary to cover all prediction grids followed by averaging for final predictions? Supplementary materials illustrating the detailed procedures of the proposed model would be valuable.*

**Response to Reviewer Comment 3:**

First, the grid cells generated by a single quadtree subdivision may not fully cover the entire study area and may omit stations located near the domain boundaries. To address this limitation, we employ six rotated quadtree configurations at different orientation angles, which collectively ensure complete spatial coverage and prevent potential loss of edge-area observations caused by a single subdivision. Second, a full 360° rotation

is unnecessary. We selected six representative angles—0°, 15°, 30°, 45°, 60°, and 75°—which sufficiently cover different directional alignments; additional angles would only introduce redundancy without improving performance. Third, we average the predictions obtained from the six rotation angles, which allows the model to capture spatial heterogeneity from multiple directional perspectives while effectively mitigating boundary artifacts induced by any single quadtree partition. This ensemble approach markedly enhances the stability and robustness of the final soil temperature estimates. Finally, following your recommendation, we have added detailed workflow diagrams and supplementary materials that illustrate the complete rotated-quadtree modeling framework, including grid rotation, spatial subdivision, model training, and prediction integration.

**Revised Text (L306-L336):**

A quadtree is a hierarchical spatial data structure that recursively subdivides a two-dimensional space into four quadrants, enabling efficient spatial indexing and localized data organization. In this study, we adopted a bottom-up, rotated quadtree–based spatial partitioning strategy that adaptively generates finer grids in regions with dense observations and coarser grids in sparsely sampled areas. Compared with global modeling or static grid partitioning, this adaptive approach improves regional modeling fidelity while maintaining computational efficiency. The procedure consists of the following steps:

(1) Initialization of Minimum Units

The entire study area was first divided into uniform minimum-sized units (leaf nodes), each representing a basic spatial element that may contain zero or more soil temperature observations. This initialization provides the base spatial resolution for subsequent hierarchical construction. An illustration of the quadtree structure and spatial indexing principles is provided in Fig. S2.

(2) Bottom-up Hierarchical Merging

Starting from the leaf nodes, groups of four adjacent quadrants were recursively merged into parent nodes if each contained fewer than 30 observation sites (threshold selection detailed in Fig. S4). The merging process continued upward until no further groups met the threshold. This approach ensures that each node has sufficient sample size while achieving spatially adaptive partitioning across the study area. Each subregion is then assigned a localized $T_s$ prediction model.

(3) Rotation at Multiple Angles

To reduce potential edge effects introduced by static grid boundaries, we implemented a rotated quadtree partitioning strategy. The quadtree structure was rotated at six angles (0°, 15°, 30°, 45°, 60°, and 75°), producing distinct sets of spatial partitions for each orientation (Fig. 3). Independent models were trained for each rotated configuration, and the final $T_s$ estimates were obtained by averaging the outputs from all six models. This rotation-based ensemble method improves spatial smoothness and minimizes discontinuities at partition boundaries.

[Figure]

**Figure 3.** Multi-angle adaptive quadtree partitioning of site observations (0°, 15°, 30°, 45°, 60°, 75°)

**Reviewer Comment 4:**

*It is recommended to add a section of pseudocode to illustrate the computational process of spatial adaptive partition method.*

**Response to Reviewer Comment 4:**

Thank you very much for your suggestion regarding improving the transparency and reproducibility of our method. In response, we have made the complete R implementation of the rotated-quadtree spatial adaptive partitioning algorithm publicly available on GitHub. The repository includes all scripts used to construct the six rotated quadtree partitions, generate the spatial blocks, and export the polygon shapefiles. The code is openly accessible at: https://github.com/wangxt1314/Rotated-quadtree

This repository also provides detailed documentation and example files, enabling users to fully reproduce the quadtree construction and subsequent modeling workflow. We believe that making the full code publicly accessible will substantially enhance the reproducibility and transparency of our study.

**Revised Text (L730-L732):**

**7. Code availability**

The R scripts used to implement the rotated-quadtree spatial adaptive partitioning are publicly available at: https://github.com/wangxt1314/Rotated-quadtree

**Reviewer Comment 5:**

*Given the simultaneous inclusion of elevation and slope (often derived from elevation) as auxiliary variables, potential multi-collinearity concerns should be addressed. Please discuss whether this correlation might affect model predictive performance.*

**Response to Reviewer Comment 5:**

Thank you very much for raising this important point. As slope is derived from the digital elevation model (DEM), it is indeed correlated with elevation. To assess whether

this relationship may introduce multicollinearity issues in our modeling framework, we conducted a Variance Inflation Factor (VIF) analysis for all auxiliary variables. The results are presented in Figure S2. Variance Inflation Factor (VIF) of predictor variables in the Supplementary Materials. The VIF values for both elevation and slope are well below commonly accepted thresholds (VIF < 10), indicating that their correlation is not strong enough to compromise model stability. Furthermore, our modeling framework is based on a tree-based algorithm (XGBoost), which learns through recursive partitioning driven by information gain. Such models are inherently robust to correlations among predictor variables and are far less susceptible to multicollinearity issues than linear regression models, where parameter estimates can become unstable under collinearity.

[Figure]

**Figure S2.** Variance Inflation Factor (VIF) of predictor variables

**Reviewer Comment 6:**
*As model predictions and uncertainty typically coexist, and given the high accuracy demonstrated in validation results, provision of corresponding prediction uncertainty estimates would strengthen the methodological rigor.*
**Response to Reviewer Comment 6:**
Thank you very much for your valuable comment. We agree that incorporating a prediction uncertainty assessment further strengthens the scientific rigor of our methodology. In response to your suggestion, we have added an uncertainty analysis based on an ensemble of six quadtree models constructed under different rotation angles.

Specifically, each observation station is predicted six times using quadtree partitioning structures generated at six different rotation angles. We quantify prediction uncertainty as the standard deviation of these six predicted $T_s$ values, which reflects the stability

and sensitivity of the model predictions to changes in spatial partitioning orientation. A larger standard deviation indicates substantial divergence among predictions from different rotations, suggesting higher structural uncertainty caused by spatial heterogeneity, partition boundary effects, or variation in local sample density. Conversely, when the six predictions remain highly consistent, the standard deviation is small, indicating that the model is stable across different partitioning orientations and exhibits lower uncertainty.

To derive a more robust annual uncertainty estimate, we further average the daily uncertainty values for each station within each year, resulting in a station-level annual uncertainty index (Fig.S14). This uncertainty metric serves as a useful complement to the prediction results by identifying areas where the model exhibits stronger structural variability or weaker observational support, thereby enhancing the transparency and credibility of the modeling results.

[Figure]

**Figure S14.** Spatial patterns of prediction uncertainty at six soil depths based on the rotated-quadtree ensemble.

The station-based uncertainty analysis shows that at the 0 cm depth, a substantially larger proportion of stations exhibit high uncertainty compared with other depths. The stations with higher uncertainty are mainly concentrated in the Sichuan Basin, the Yunnan–Guizhou Plateau, and the Qinghai–Tibet Plateau, which are characterized by complex geological and geomorphological environments. In contrast, the overall uncertainty levels at the remaining depths are considerably lower and spatially more stable. We believe that incorporating this improvement will further strengthen the

methodological rigor, enhance the reliability of the results, and provide valuable guidance for future users of the dataset.

---

## Author Response (AR3)

**Revisions of Manuscript:  ESSD-2025-192**

**Title:** Spatially adaptive estimation of multi-layer soil temperature at a daily time-step across China during 2010-2020

**Author(s):** Xuetong Wang, Liang He, Peng Li, Jiageng Ma, Yu Shi, Qi Tian, Gang Zhao, Jianqiang He, Hao Feng, Hao Shi, Qiang Yu

Dear Reviewer,

We sincerely thank you for your thoughtful comments and constructive suggestions on our manuscript. We have carefully revised the manuscript in response to your feedback, with all changes clearly marked using track changes. In the revised manuscript and accompanying supplementary materials, modifications are highlighted in blue for ease of reference.

Below, we provide a detailed, point-by-point response to each of your comments. For clarity, your original remarks are shown in *italics*, followed by our corresponding replies. We have made every effort to address all concerns comprehensively and to improve the scientific rigor, clarity, and overall quality of the manuscript.

We sincerely appreciate the time and effort you invested in reviewing our work.

**Reviewer Comment 1:**

*Figures S12, S13, and S14 in the Supplementary are not referenced anywhere in the main text. Please ensure all supplementary figures are explicitly cited in the relevant sections of the manuscript.*

**Response to Reviewer Comment 1:**

We thank the reviewer for pointing out the missing references to Figures S12, S13, and S14 in the main text. To address this, we have now explicitly cited these figures in the relevant sections of the manuscript. Specifically:

(1) Figure S12 has been updated to Figure S11 and is referenced in L438-442 of the manuscript.

**Revised Text (L438-L442):**

Figure S11 shows the comparison between the estimated and observed annual mean $T_s$ for the test dataset at six different depths (0~40 cm). The R² ranges from 0.94 to 0.97. The RMSE values range from 0.74 to 1.4 K, and the bias is minimal. The results suggest that the model is able to effectively capture the spatial patterns of $T_s$ across different depths and locations.

(2) Figure S13 has been updated to Figure S9 and is referenced in L419-423 of the manuscript.

**Revised Text (L419-L423):**

In Figure S9, we further validated the spatial consistency between the flux tower sites and the estimated annual mean $T_s$ at different depths. Although the validation results demonstrated high accuracy overall (R² = 0.7~0.82; RMSE = 2.93~3.58 K), a systematic positive bias of approximately +2 to +3 K was observed across all depths.

(3) The content related to Figure S14 has been referenced in L587-591 of the manuscript.

**Revised Text (L587-L591):**

We also quantified the variability of prediction results at the same site using grids generated from different rotation angles. The results in Fig. S14 show that the uncertainty at the 0 cm depth is higher compared to other depths, with the highest uncertainty concentrated in certain areas of the YGP and Sichuan Basin.

**Reviewer Comment 2:**

*I welcome the addition of Figure S13, which compares the annual mean Ts against the independent flux tower observations. The R² values reduce to 0.70–0.87, which are acceptable, but there is a consistent and significant positive bias of +2 to +3 K across all depths. A systematic bias of ~3 K is not trivial for soil temperature applications. At least, a brief acknowledgement of this bias should be included in Section 3.1 (where flux tower validation is discussed).*

**Response to Reviewer Comment 2:**

We appreciate the reviewer's constructive comment regarding the comparison between

Figure S13 and independent flux tower observations. We acknowledge that, although the results are still acceptable across all depths, with R² values ranging from 0.70 to 0.82, there is indeed a consistent and significant positive bias of approximately +2 to +3 K. We recognize the importance of highlighting this bias in the main text and have made the necessary revisions.

To address this comment, we have changed Figure S13 in the original supplementary materials to Figure S9, and we have added the corresponding explanation in Section 3.1 of the main text.

**Revised Text (L419-L423):**
In Figure S9, we further validated the spatial consistency between the flux tower sites and the estimated annual mean $T_s$ at different depths. Although the validation results demonstrated high accuracy overall (R² = 0.7~0.82; RMSE = 2.93~3.58 K), a systematic positive bias of approximately +2 to +3 K was observed across all depths.

**Reviewer Comment 3:**
*In the caption of Figure S14, please explicitly state that this uncertainty metric represents only the variability induced by the spatial partitioning scheme, rather than the total predictive uncertainty.*

**Response to Reviewer Comment 3:**
We thank the reviewer for the insightful suggestion. We have revised the caption of Figure S14 to explicitly clarify that the uncertainty metric shown in the figure represents only the variability induced by the spatial partitioning scheme, rather than the total predictive uncertainty.

**Here are the revisions, supplemented in the Appendix (L122-L129):**

[Figure]

**Figure S14.** Spatial patterns of prediction uncertainty at six depths (0~40 cm) based on the rotated-quadtree ensemble. Note:The uncertainty metric shown here represents the variability induced by the spatial partitioning scheme rather than the total predictive uncertainty. Colored points represent site-level uncertainty values, with warmer colors indicating higher uncertainty. Insets show the frequency distribution histograms of uncertainty levels at each depth.

We believe this revision adequately addresses the comment and enhances the clarity of the figure caption.